



# Carbon degradation in agricultural soils flooded with seawater after managed coastal realignment

Kamilla S. Sjøgaard[1], Alexander H. Treusch[1] and Thomas B. Valdemarsen[1]

[1]Department of Biology, University of Southern Denmark, Odense M, 5230, Denmark

*Correspondence to*: Kamilla S. Sjøgaard (kamillasjogaard@biology.sdu.dk)

**Abstract.** Climate change induced sea level rise is expected to continue for centuries and cause permanent flooding of low lying coastal areas. Furthermore, intentional flooding of coastal areas through 'managed coastal realignment', may also become a common solution to protect coastal areas. So far, the biogeochemical implications of flooding soils with seawater are not well investigated. In this study we conducted a 1-year mesocosm experiment to investigate microbial carbon

degradation processes in soils flooded with seawater. The used soils were sampled at on Northern Fyn (Denmark), in an area (Gyldensteen Strand) that was planned to be flooded in a coastal realignment project. We found rapid carbon degradation almost immediately after flooding and microbial sulfate reduction rapidly established as the dominant mineralization pathway. Nevertheless, no free sulfide was observed as it precipitated as Fe-S compounds with Fe acting as a natural buffer, preventing toxic effects of free sulfide in soils flooded with seawater. The refractory nature of the terrestrial organic carbon

in combination with the anoxic conditions created in the soil after flooding caused significantly decreased organic carbon degradation after 6 months. During the experiment only 6-7 % of the initial organic carbon pools were degraded. On this basis we suggest that flooding of coastal soils through sea level rise or managed coastal realignment, will cause significant C-preservation and create a negative feedback on atmospheric carbon dioxide concentrations.

## 1 Introduction

Sea level rise driven by global climate change is expected to continue for centuries. Modelled scenarios in the Fifth Assessment Report (AR5) from 2013 the Intergovernmental Panel on Climate Change (IPCC) on global mean sea level (GMSL) rise suggest a GMSL rise between 0.26-0.82 m during the 21st century. Based on long-term scenarios it is almost certain that GMSL rise will continue beyond 2100 primarily due to thermal water expansion. About 70 % of the global coastlines are forecast to be impacted by a sea level rise within 20 % of the GMSL change (Church et al., 2013).

Rising sea level causes higher and more frequent storm surges, which lead to more incidences of floodwaters or storm surges overtopping and breaking coastal defenses (FitzGerald et al., 2008). Reclaimed coastal areas with low elevation are especially vulnerable to flooding by storm surges. A low cost coastal defense strategy is 'managed coastal realignment', whereby old coastal defenses are deliberately breached, and new ones are constructed further inland (Cooper, 2003; French, 2008). The flooded areas created by managed coastal realignment act as buffer zones protecting populated areas or valuable





assets against flooding. This technique has also been used for nature restoration of marshlands (Pethick, 2002; Wolters et al., 2005). The number of managed coastal realignment projects will increase as sea level rise intensifies, so it is important to know the ecological implications of introducing seawater into soil and freshwater ecosystems.

We expect that flooding with seawater will have dramatic consequences for soil biogeochemistry. Depending on

soil porosity and moisture content, soil environments can have deep oxygen penetration (75-100 cm) (Dziejowski et al., 1997; MacDonald et al., 1993; Neira et al., 2015), since oxygen ($O_2$) can rapidly be supplied from the overlying atmosphere via diffusion. Therefore, surface soils are predominantly oxic environments where soil organic matter is degraded by aerobic processes performed by a wide variety of bacteria, fungi and fauna (Boer et al., 2005; Kalbitz et al., 2000). During aerobic degradation reactive oxygen radicals are formed that can break bonds in refractory organic compounds such as lignin and

cellulose (Canfield, 1994). However, when soils are flooded, $O_2$ penetration will be dramatically reduced, since $O_2$ solubility in water is low and $O_2$ diffusion in water is $10^4$ times slower than in air (Neira et al., 2015). $O_2$ will therefore be depleted by microbial and chemical $O_2$ consuming processes in surface soils, which become anoxic below a few mm depth. In anoxic environments mutualistic consortia of microorganisms degrade organic macromolecules into smaller moieties by excretion of exoenzymes and extracellular hydrolysis, which are then fermented into smaller organic molecules, mainly acetate

(Valdemarsen and Kristensen, 2010). The fermentation products are taken up by microorganisms and terminally oxidized to carbon dioxide ($CO_2$) by the consumption of alternative electron acceptors (e.g. nitrate, Mn oxides, Fe oxides and sulfate) (Arnosti, 2011; Glud, 2008). In soils flooded with seawater, microbial sulfate reduction (SR) is expected to become a major pathway for organic matter degradation (Weston et al. 2011), but it is uncertain how fast SR can establish here. Further it is unclear to what extent soil organic matter can be degraded and serve as substrate for anaerobic microorganisms typical for

marine environments.

Many soils subject to managed coastal realignment often contain considerable amounts of soil organic carbon (OC) (Franzluebbers, 2010; Wolters et al., 2005). The degradation of soil OC after flooding will depend on the rate of establishment of heterotrophic microbial communities and their ability to degrade soil OC (Schmidt et al., 2011). Labile OC may be easily degraded by marine microorganisms, while more complex OC, and especially structurally complex OC

consisting of cellulose and lignin, may be virtually non-degradable in anoxic environments (Kim and Singh, 2000; Kristensen and Holmer, 2001). Flooding of coastal soils due to GMSL rise and coastal realignment may therefore cause significant preservation of soil OC, leading to negative feedbacks on atmospheric $CO_2$ concentrations.

In this study the fate of soil OC after flooding with seawater was investigated in soils collected at Gyldensteen Strand on Northern Fyn, Denmark. The sampled area was designated to be flooded in a coastal realignment project. We were

especially interested in following the temporal establishment of dominating microbial pathways and quantifying the rates and temporal trajectories of soil OC degradation in newly flooded soils. We hypothesized that (1) total OC degradation activity in soils after flooding depends on soil OC content and lability, and that (2) most soil OC at the time of flooding will, due to it's terrestrial origin, be non-degradable and hence preserved under the anoxic conditions formed after the flooding. Therefore it is plausible that (3) the majority of soil OC will be preserved in flooded soils creating a negative feedback on





atmospheric $CO_2$ concentrations. To test these hypotheses we performed parallel mesocosm experiments with two different types of soils that were experimentally flooded with seawater. OC-degradation and other biogeochemical developments in the flooded soils were then followed with high temporal and spatial resolution for the next 12 months. The results showed how flooding with seawater impacts C-degradation and soil biogeochemistry, and formed the basis for an initial evaluation

of potential feedbacks of flooding on atmospheric $CO_2$ concentrations.

## 2 Materials and methods

### 2.1 Study site

This study was conducted in relation to the nature restoration project at Gyldensteen Strand funded by the Danish Aage V. Jensen Nature Foundation. The sampling site (55°34'26.4"N 10°08'17.0"E) was a shallow intertidal habitat until 1871 (size

of ~600 ha), where it was reclaimed to create new land for agriculture. The reclaimed area was for the following 140 years mainly used for production of different crops such as onions and grains (Stenak, 2005). As a part of the nature restoration project, selected sections of the dikes were removed in March 2014 and 211 ha of the area were permanently flooded with seawater and turned into a shallow marine lagoon.

### 2.2 Experimental design and Sampling

Sampling for the mesocosm experiment was performed in November 2013, half a year before the flooding, at two different stations representing uncultivated (UC) and cultivated (C) soils (Fig. 1). Station UC was located in an area with low elevation, and it could never be properly drained. Station UC was therefore abandoned for agriculture and became a reed swamp that accumulated plant material and litter. Station C however, was similar to the majority of the re-flooded area that was farmed since the land reclamation (fertilized, ploughed and used for monoculture). From each station, 15 soil cores were

sampled in 30 cm long, 8 cm internal diameter stainless steel core liners. The core liners were hammered 25 cm down into the soil, dug up with a spade and closed in both ends with rubber stoppers.

In the laboratory, the headspaces of individual soil cores were gently flooded with 22-26 salinity seawater collected from the shore face directly North of station UC (Fig. 1). Soil cores were then transferred to 70 L incubation tanks filled with seawater. During the whole experiment the flooded cores were maintained at 15 °C and kept in darkness. The water in the

tanks was rigorously aerated through air diffuser stones and 10-20 L of the seawater in the tanks was exchanged with fresh seawater every 14 days.

The flooded soil cores were incubated for 12 months. Flux experiments were conducted with 3 random soil cores from each station at various times (weekly in the first month, biweekly for the next 3 months and monthly hereafter). Core sectionings were performed on 3 randomly selected soil cores from each station at different times during the experiment

(before the flooding, 1 week after and after 2, 4, 6 and 12 months).





### 2.2.1 Flux experiments

Fluxes of $O_2$, DOC and $TCO_2$ (= $CO_3^{2-}$ + $HCO_3^-$ + $H_2CO_3$) between soil and overlying water were measured regularly as described above. Cores were equipped with stirring magnets, closed with rubber stoppers and placed around a central magnet rotating at 60 rpm and hereafter incubated for about 4 hours in darkness. $O_2$ was measured and water samples were taken in

the headspace of the soil cores at the beginning and end of incubations. $O_2$ was measured with an optical dissolved oxygen meter (YSI ProODO). DOC samples were stored at -20 °C until analysis using a Shimadzu TOC-5000 Total Organic Analyzer. Samples for $TCO_2$ analysis were kept in 3 mL gas-tight exetainers for maximum 1 week until analyzed by flow injection (Hall and Aller, 1992).

### 2.2.2 Core sectioning

Core sectioning was performed by slicing each soil core into 6 depth intervals (0-1, 1-2, 3-5, 5-10, 10-15 and 15-20 cm). Porewater was extracted from each depth interval by centrifugation and GF/C filtration in double centrifuge tubes (500 g, 10 min). The porewater was sampled for various parameters; 500 μL porewater was preserved with 30 μL saturated $HgCl_2$ for $TCO_2$, 250 μL porewater was preserved with 50 μL 1 M zinc acetate (ZnAc) for total dissolved sulfide ($TH_2S$ = $H_2S$ + $HS^-$ + $S_2^-$) analysis, 250 μL porewater was preserved with 100 μL 0.5 M HCl for $Fe^{2+}$ analysis and remaining porewater was stored

at -20 °C until analysis for sulfate ($SO_4^{2-}$) and dissolved organic carbon (DOC). $TCO_2$ and DOC samples were stored and analyzed as described above. $TH_2S$ samples were analyzed by the method of Cline (1969). $Fe^{2+}$ samples were analyzed by the Ferrozine method (Stookey, 1970). $SO_4^{2-}$ was analyzed by liquid ion chromatography on a Dionex ICS-2000 system.

Reactive iron, RFe, was extracted from soil subsamples from every depth interval with 0.5 M HCl for 30 min while shaking (Lovley and Phillips, 1987). After centrifugation (500 g, 10 min) the supernatant was transferred to sampling vials

and stored at room temperature until analysis for reactive Fe(II) and Fe(III) [RFe(II) and RFe(III), respectively]. The supernatant was analyzed for $Fe^{2+}$ and RFe by the ferrozine method (Stookey 1970) before and after reduction with hydroxylamine (Lovley and Phillips, 1987). RFe(II) was calculated directly, while RFe(III) was calculated from the difference between RFe and RFe(II). An estimate of total Fe content was obtained by boiling combusted soil subsamples in 1 M HCl for 1 hour at 120 °C. The supernatant was stored at room temperature until analysis by the ferrozine method.

Acid volatile sulfides (AVS) (Rickard and Morse, 2005) and chromium reducible sulfur (CRS) was determined on soil subsamples preserved with 1 M ZnAc and stored at -20 °C until analysis. AVS and CRS was extracted by 2-step distillation as described in Fossing and Jørgensen (1998). Sulfide concentrations in the distillates were analyzed by the method described by Cline (1969).

Soil characteristics were also determined for every depth interval during every core sectioning. Soil density was

determined gravimetrically and soil subsamples were dried (24 h, 105 °C) to determine water content and porosity. Soil organic matter content was measured as the weight loss of dry sediment after combustion (520 °C, 5 hours). Organic carbon





(OC) on selected soil samples (samples obtained after 1 week and 6 months) was also measured by elemental analysis on Carlo Erba CHN EA1108 Elemental Analyzer according to Kristensen and Andersen (1987).

### 2.2.3 Anoxic incubations (Jar experiments)

Depth distribution of microbial $TCO_2$ and DOC production and SR were estimated from anoxic soil incubations (Kristensen
and Hansen, 1995; Quintana et al., 2013). The excess soil from core sectionings was pooled into 4 depth intervals (0-2, 2-5, 5-10 and 15-20 cm), thoroughly homogenized and tightly packed into 6-8 glass scintillation vials (20 mL). The vials were closed with screw caps, buried head-down in anoxic mud and incubated at 15 °C in darkness. 2 jars from each jar series were sacrificed every week for porewater extraction in the following 4 weeks. The screw caps were changed to a perforated lid containing a GF/C filter and the jars were centrifuged upside-down in a centrifuge tube (10 min at 500 g). The extracted
porewater was sampled and analyzed for $TCO_2$, DOC and $SO_4^{2-}$ as described above.

### 2.3 Data analysis

Fluxes of $TCO_2$, DOC and $O_2$ were calculated from the concentration difference between start and end samples. Microbial rates in jar experiments (DOC and $TCO_2$ production and SR) were calculated for 0-2, 2-5, 5-10, 15-20 cm depth intervals by fitting the time dependent concentration changes by linear regressions. When the slopes were significant ($p < 0.05$), the
volume specific reaction rates (nmol cm$^{-3}$ d$^{-1}$) in individual depth layers were calculated from the regression slopes corrected for sediment porosity. Microbial reaction rates, porewater and solid pools were depth integrated over 0-20 cm and converted to area specific units. Linear data interpolation was used to correct for missing data points, e.g. for the depth interval 10-15 cm where microbial rates were not measured. There was a significant linear correlation between organic matter content and OC for both sampling stations. This correlation was used to convert organic matter into OC for the time points where OC
was not directly measured. A one-way ANOVA was performed on OC (%) at the different times to test for significant changes in OC pools over time. Depth integrated SR rates were normalized to C-units assuming a 2:1 ratio between $CO_2$ production and SR. For soil characteristics, fluxes, porewater and solid pools, error was calculated as standard error of the mean (SEM). For depth-integrated values of microbial rates and solid pools error was calculated as standard error propagation (SEP) of standard deviation (SD) values following Eq. (1):

$$SEP = \sqrt{SD_{0-1\,cm}^2 + \cdots + SD_{15-20\,cm}^2} \qquad (1)$$

In the carbon budget, the total OC degradation was calculated as the sum of the time integrated $TCO_2$ efflux, time integrated DOC efflux and area specific $TCO_2$ and DOC in porewater by the end of the experiment (mol m$^{-2}$).



## 3 Results

### 3.1 Soil characteristics

The two sampled stations had very different soil appearance, as a result of different use after the land reclamation (i.e. no cultivation and cultivation). Station UC was overgrown with mosses and grasses, and a dense layer of roots and litter
characterized the upper 5 cm of the soil, while the deeper parts of the soil (>10 cm depth) consisted of clay. On station C only relatively small amounts of grass and root material were evident in the upper 5 cm. Some of the vegetation was still alive 2 months after the flooding, as indicated by long green grass leaves seeking light, but it slowly died out thereafter. The soil at both stations contained partially degraded shell material from gastropods and bivalves remaining from when the area was a marine lagoon before 1871.

There was very little variation in soil characteristics between successive core sectionings, so results were averaged for the whole experiment (Table 1). The water content at station UC decreased with depth from 83 % at the top to 35 % in the bottom, while water content only decreased from 32 % to 20 % at station C. The same depth-trend was observed for porosity. The high water content and porosity at station UC was caused by high amounts of plant material (e.g. roots) while the soil at station C was sandy, homogenous and poor in organic debris.

Soil organic content varied greatly with depth at station UC, and the topsoil was enriched with OC (16 %) compared to the bottom (1 %) (Table 1). Soil OC varied between 0.8 and 1.4 % at station C and there was no depth variation. A one-way ANOVA showed no significant difference between the OC contents at the different time points at either station UC or C (df = 17, F = 1.9, p = 1.16 for both stations).

### 3.2 $CO_2$ and DOC efflux, and $O_2$ consumption

$TCO_2$ effluxes in UC soil were highest in the beginning of the experiment with a maximum of 239±30 mmol m$^{-2}$ d$^{-1}$ measured on day 13 (Fig. 2a). Then it decreased to about 130 mmol m$^{-2}$ d$^{-1}$ 31-199 days after flooding and stabilized around 67 mmol m$^{-2}$ d$^{-1}$ from day 220 to the end. The $TCO_2$ effluxes in C soil were relatively constant around an average of 29 mmol m$^{-2}$ d$^{-1}$.

       High DOC efflux was evident 1 day after flooding at station UC (108±3 mmol m$^{-2}$ d$^{-1}$) (Fig. 2b), while it decreased
to around 60 mmol m$^{-2}$ d$^{-1}$ 6-20 days after flooding and to 17 mmol m$^{-2}$ d$^{-1}$ after approximately 2 months to the end. DOC effluxes at station C showed a similar pattern, averaging 25 mmol m$^{-2}$ d$^{-1}$ in the first 2 months after flooding, and decreasing to an average of 5 mmol m$^{-2}$ d$^{-1}$ for the remaining experiment.

       $O_2$ consumption decreased almost linearly during the 1-year experiment on both stations (Fig. 2c). At station UC initial $O_2$ consumption was 57±3 mmol m$^{-2}$ d$^{-1}$, 1-45 days after flooding, and then it steadily decreased to 19±3 mmol m$^{-2}$ d$^{-1}$
by the end. At station C there was a less pronounced temporally decreasing trend. $O_2$ consumption was highest initially with about 26 mmol m$^{-2}$ d$^{-1}$ at day 1-13 and then decreased to 9±0.6 mmol m$^{-2}$ d$^{-1}$ by the end.



### 3.3 Porewater chemistry

Porewater DOC was high 1 week after flooding on both stations (on average 10.4 and 3.8 mM on station UC and C, respectively; Fig. 3a). Over the experiment porewater DOC decreased slightly in UC soil, while it increased slightly in C soil.

Porewater $TCO_2$ concentrations in UC soil were in the range of 5-13 mM between 1 week and 2 months after flooding, and profiles showed a slightly increasing pattern with depth (Fig. 3b). Afterwards an unexpected drop in $TCO_2$ concentrations, especially in the deep soil (>2 cm depth), was observed. This was likely an experimental artifact, however, caused by extremely high $Fe^{2+}$ concentrations >2 mM in the porewater. During sample storage the $Fe^{2+}$ got oxidized to Fe-oxyhydroxides and formed an orange-brown precipitate at the bottom of sample containers, and probably led to sample-

acidification and $TCO_2$ degassing (Moses et al. 1987; Hedin 2006). Porewater $TCO_2$ concentrations in UC soil after 4 months were affected by this artifact. In C soil, porewater $Fe^{2+}$ did not accumulate at the same rate as in UC soil, and only exceeded 2 mM in the 10-20 cm depth layer after 6 months, and porewater $TCO_2$ accumulated gradually over time as expected (Fig. 3b). Rapid $TCO_2$ accumulation occurred in the first 2 months, where $TCO_2$ increased from 3-5 mM to 11 mM below 3 cm depth. After 2 months to the end, $TCO_2$ increased further in the 2-10 cm depth interval, while a decrease

occurred below 10 cm depth, which was probably related to $Fe^{2+}$ exceeding 2 mM.

High concentrations of $SO_4^{2-}$ were introduced to the soil when flooded with seawater. Yet diffusion was the only transport mechanism for dissolved $SO_4^{2-}$ in the mesocosm setup, and the experimental period was evidently not sufficiently long to achieve full saturation of $SO_4^{2-}$ in porewater down to 20 cm depth. As a result, porewater concentrations of $SO_4^{2-}$ decreased steeply with depth at both stations (Fig. 3c). By the end of the experiment in UC soil, $SO_4^{2-}$ decreased from ~17

mM at the surface to zero below 10 cm depth. In C soil $SO_4^{2-}$ decreased linearly from ~17 mM at the surface to 0-2 mM at the bottom.

After 7 days of flooding the $Fe^{2+}$ depth distribution in porewater was constant with depth, with on average 0.02 and 0.2 mM at station UC and C, respectively (Fig. 3d). Afterwards a progressive increase in porewater $Fe^{2+}$ was observed at both stations. At station UC $Fe^{2+}$ increased to up to 1.3±0.6 mM at 5-15 cm depth after 2 months and stabilized after 6

25   months, where $Fe^{2+}$ exceeded 4 mM below 5 cm depth. The same trend was seen at station C, where $Fe^{2+}$ accumulated to up to 3.7 mM at 15-20 cm depth after 12 months.

### 3.4 Anaerobic net DOC production in jar experiments

Net DOC production after 1 week of flooding was high in the surface 0-2 cm at station UC (2666±695 nmol cm$^{-3}$ d$^{-1}$; Fig. 4a) and decreased exponentially with depth to 203±23 nmol cm$^{-3}$ d$^{-1}$ at 15-20 cm depth. A gradually decreasing net DOC

production was observed in all depth layers over the experiment, and by the end, significant net DOC production (121-172 nmol cm$^{-3}$ d$^{-1}$) was only detected in the upper 0-5 cm. A similar pattern in net DOC production was observed at station C, although rates were much lower than at station UC. After 1 week of flooding, net DOC production at station C was





$1155\pm158$ nmol $cm^{-3}$ $d^{-1}$ in the upper 0-2 cm of the soil, but only 66-83 nmol $cm^{-3}$ $d^{-1}$ below. After 4 months it had decreased to 135 nmol $cm^{-3}$ $d^{-1}$ in the top 0-2 cm and no net DOC production was detected below 5 cm depth. Very low rates (21-25 nmol $cm^{-3}$ $d^{-1}$) were detected in the top 0-5 cm by the end.

Depth integrated net DOC production at station UC was initially 118-133 mmol $m^{-2}$ $d^{-1}$ in the first 2 months after flooding and then gradually declined to 8 mmol $m^{-2}$ $d^{-1}$ after 12 months (Fig. 5). Initial depth integrated net DOC production at station C was 4-fold lower than at station UC. Net DOC production in C soil decreased by 75 % in the first 2 months after flooding and almost no net DOC production occurred after 6 months.

### 3.5 Anaerobic TCO$_2$ production in jar experiments

Initial depth trends in $TCO_2$ production were generally similar to those observed for DOC, but temporal trends were markedly different (Fig. 4b). At station UC, $TCO_2$ production was initially almost 1000 nmol $cm^{-3}$ $d^{-1}$ in the top 0-2 cm and decreased to 380 nmol $cm^{-3}$ $d^{-1}$ at 15-20 cm depth. After 2 months, $TCO_2$ production had increased in the surface 0-2 cm to 6250 nmol $cm^{-3}$ $d^{-1}$, while rates below 10 cm depth remained relatively low. After 4 months, $TCO_2$ production decreased to about 2500 nmol $cm^{-3}$ $d^{-1}$ in the top 0-2 cm, while it was not possible to determine $TCO_2$ production rates directly for soil deeper than 5 cm, due to the problem with extremely high porewater $Fe^{2+}$ described above. As seen below, porewater $SO_4^{2-}$ concentrations were not affected by the high porewater $Fe^{2+}$ concentrations. So for the affected data points, $TCO_2$ production was calculated as rate of SR x 2, assuming that SR was the dominating $CO_2$ producing process in the anoxic soil. The calculations showed that $TCO_2$ production had decreased further after 6 and 12 months in the top 5 cm (600-1000 nmol $cm^{-3}$ $d^{-1}$) and was quite stable below (0-85 nmol $cm^{-3}$ $d^{-1}$). $TCO_2$ production rates were generally much lower in C soil, while relative trends for $TCO_2$ production and their development over time were quite similar between stations. Maximum $TCO_2$ production rates occurred at 0-2 cm depth, where $TCO_2$ production varied from 400 to 780 nmol $cm^{-3}$ $d^{-1}$ between 1 week and 2 months and then gradually decreased to 110 nmol $cm^{-3}$ $d^{-1}$ by the end. Similar trends were observed in the deeper soil where $TCO_2$ production decreased from 180-310 nmol $cm^{-3}$ $d^{-1}$ after 7 days to 7-53 nmol $cm^{-3}$ $d^{-1}$ after 12 months.

Area specific $TCO_2$ production at station UC was initially 115-200 mmol $m^{-2}$ $d^{-1}$ in the first 2 months, and decreased to 40 mmol $m^{-2}$ $d^{-1}$ after 6 months (Fig. 5). At station C area specific $TCO_2$ production was relatively stable around 44 mmol $m^{-2}$ $d^{-1}$ for the first 4 months and was decreased to 21 and 10 mmol $m^{-2}$ $d^{-1}$ after 6 and 12 months, respectively.

### 3.6 SR in jar experiments

Significant SR was measured in the top 0-5 cm (470 nmol $cm^{-3}$ $d^{-1}$) in UC soil after 1 week of flooding, while no SR was detected below (Fig. 4c). After 2 months, high SR was only measured in the top 0-2 cm ($3128\pm190$ nmol $cm^{-3}$ $d^{-1}$). After 4 months SR was still highest in the topsoil ($1217\pm147$ nmol $cm^{-3}$ $d^{-1}$), while significant SR was detected down to 10 cm depth. From 4 months to the end, SR gradually decreased at all depths to $338\pm147$ and $43\pm6$ nmol $cm^{-3}$ $d^{-1}$ at 0-2 and 5-10 cm depth, respectively. Since $SO_4^{2-}$ did not reach the bottom (15-20 cm) during the experiment at station UC, no SR occurred here. In C soil SR occurred at considerably lower rates than in UC soil. After 1 week SR was $177\pm25$ nmol $cm^{-3}$ $d^{-1}$ at 0-2 cm



depth and decreased exponentially with depth to zero at 15-20 cm depth. By month 2 and 4, SR occurred at all depths (20-159 nmol cm$^{-3}$ d$^{-1}$). Afterwards SR decreased in the upper 15 cm while no SR was detected in the 15-20 cm depth interval.

Depth integrated SR at station UC increased from 24 to 63 mmol m$^{-2}$ d$^{-1}$ between week 1 and month 2, corresponding to 48 and 126 mmol m$^{-2}$ d$^{-1}$ carbon mineralization, respectively (Fig. 5). SR had decreased to 27.7 mmol m$^{-2}$ d$^{-1}$ after 12 months. SR increased during the first 4 months in C soil (6 to 12 mmol m$^{-2}$ d$^{-1}$) and then decreased to 4 mmol m$^{-2}$ d$^{-1}$ after 12 months.

### 3.7 Solid pools of Fe and S

Before flooding, RFe(II) in UC soil increased with depth from 4 μmol cm$^{-3}$ at 0-1 cm depth to 13 μmol cm$^{-3}$ at 15-20 cm depth, while a corresponding increase in RFe(III) occurred from 19 to 44 μmol cm$^{-3}$ (Fig. 6). The RFe pools at station C were relatively constant with depth, on average 2.5 and 23 μmol cm$^{-3}$ for RFe(II) and RFe(III), respectively. 12 months after flooding, RFe(II) in UC soil had increased to 34-59 μmol cm$^{-3}$, while RFe(III) had accumulated to 134.5±85 μmol cm$^{-3}$ in the top and decreased to an average of 4 μmol cm$^{-3}$ below. A similar trend was obtained in C soil with RFe(III) accumulating to 51.9±1.4 μmol cm$^{-3}$ on the surface. In UC and C soil, total RFe initially consisted of 78 and 92 % Fe(III), respectively, while it was reduced to 19 and 10 % by the end. Clearly, RFe(III) became reduced to RFe(II) during the experiment due to the anoxic conditions created by flooding.

The RFe content was quite heterogeneous at the study sites, and there was large variation between soil cores. Based on all the depth profiles obtained over the experiment, average total Fe content in UC and C soil was 19.3±2.8 mol m$^{-2}$ and 26.7±1.8 mol m$^{-2}$, respectively.

Although jar experiments suggested high SR in both soil types, dissolved sulfide (TH$_2$S) was never detected in the porewater. Instead, a large fraction of the sulfide produced during SR accumulated as AVS and CRS in both soil types (Fig. 7). 1 week after flooding, AVS and CRS in UC soil were low (0.2-2.7 μmol cm$^{-3}$), except at 2-5 cm depth where AVS content was slightly elevated. 12 months after flooding, AVS and CRS had increased to 25±10 and 41±11 μmol cm$^{-3}$ at 2-5 cm depth, respectively, while no accumulation occurred below 10 cm depth. A similar pattern was observed in C soil, where AVS and CRS were initially constant with depth averaging 0.1 and 3.5 μmol cm$^{-3}$, respectively, and accumulated to 6.4±1 and 8.4±0.7 μmol cm$^{-3}$ after 12 months of flooding, respectively. Over the whole experiment total sulfide accumulated as AVS and CRS gradually increased, from 0.5 mol m$^{-2}$ before flooding to 4.7 mol m$^{-2}$ after 12 months in UC soil, and from 0.63 to 2 mol m$^{-2}$ in C soil.

### 3.8 Budgets for OC degradation

Area specific OC pools were 710.9±54 and 232.5±22 mol m$^{-2}$ (n = 18) in UC and C soil, respectively (Table 2). Total OC degradation estimated as the sum of TCO$_2$ and DOC effluxes, and porewater accumulation over the 1-year experiment was 49.6 and 14.8 mol m$^{-2}$ at station UC and C, respectively, corresponding to 7 and 6 % of the OC pools.



Total OC mineralization to $TCO_2$ was estimated as the sum of $TCO_2$ efflux and porewater accumulation during the whole experiment (Table 3), which was 40.0 and 12.0 mol m$^{-2}$ at station UC and C respectively. The importance of anaerobic OC degradation for total $TCO_2$ mineralization could be calculated from jar experiments, and a total of 32.6 and 10.8 mol m$^{-2}$ OC was converted to $TCO_2$ anaerobically, corresponding to 82 and 90 % of flux-based total $TCO_2$ production at station UC and C, respectively. The SR measured in jar experiments corresponded to 25.3 and 4.3 mol m$^{-2}$ $CO_2$ production at station UC and C during the experiment. Thus 63 and 36 % of the flux-based total $TCO_2$ production was driven by SR in UC and C soil, respectively. This means that the remaining 19 and 54 % of the flux-based total $TCO_2$ production was produced by other anaerobic processes than SR in UC and C soil, respectively (e.g. nitrate or Fe reduction).

Based on the above calculations, aerobic OC degradation contributed to 18 and 10 % to the total $TCO_2$ mineralization during the 1-year experiment. However, the total $O_2$ consumption measured in the flux experiment (13.4 and 6.2 mol m$^{-2}$) was 34 and 52 % of total $TCO_2$ mineralization in UC and C soil, respectively.

## 4 Discussion

### 4.1 Temporal trends in OC degradation

The UC and C soil had very different organic content. UC soil had not been used for agriculture and organic matter consisting of dead and alive plant matter had accumulated in the topsoil (Table 1), while lower organic matter content was evident in C soil, due to lower plant cover and regular mechanical soil reworking during agricultural cultivation (Benbi et al., 2015; Six et al., 1998). Consequently, the total OC pool was 3 times higher in UC soil than in C soil. The source of soil organic matter at both stations was terrestrial plants such as grasses and herbs rich in cellulose and lignified tissues (Arndt et al., 2013; Sullivan, 1955). Such organic matter is refractory towards degradation in anaerobic marine sediments (Kristensen, 1990, 1994) compared to structurally simple phytoplankton, microphytobenthos and macroalgae, which are common OC sources in coastal marine sediments (Dubois et al., 2012; Fry et al., 1977). It was therefore uncertain to which extent the soil organic matter at Gyldensteen Strand could serve as substrate for developing microbial communities after the flooding with seawater. Nevertheless, we observed high heterotrophic activity (e.g. $O_2$ uptake and $TCO_2$ production) right after the flooding, indicating that at least part of the OC in both soil types was readily available for microbial degradation.

Cleavage of particulate OC to DOC by extracellular enzymes is the primary degradation step in waterlogged anoxic soils and sediments (Arnosti, 2011; Weiss et al., 1991). The produced DOC is hereafter converted into short chain fatty acids, mainly acetate, by microbially mediated fermentation and hydrolysis, which then are terminally oxidized to $CO_2$ by e.g. SR (Canfield et al., 2005; Valdemarsen and Kristensen, 2010). DOC production can therefore be considered the rate-limiting step for OC degradation. In this experiment we observed high DOC concentrations in porewater and highest DOC production in jar experiments already 7 days after flooding with seawater (Fig. 3a & 5). Part of this DOC may have leached to the porewater as a result of flooding (Kalbitz et al., 2000), while the rest was produced by microbial degradation of



particulate OC (Kim and Singh, 2000). Microbial degradation of soil organic matter to DOC was initiated immediately after flooding irrespective of the shift to anoxic conditions. Differences in DOC production rates indicated that the availability of degradable OC was clearly highest in UC soil compared to C soil following the overall difference in total OC content. However, total DOC production ceased rapidly in both soil types and was close to zero after 1 year. It therefore appears that

only a minor portion of soil OC (6-7 %; Table 2) is available for microbial degradation under the present conditions (flooded with seawater and anoxic conditions). The low degradability of soil OC after flooding probably reflects limitations of the anaerobic microbial communities to degrade complex organic matter of terrestrial origin.

Heterotrophic DOC oxidizing microbes were also active immediately after flooding as shown by initial $TCO_2$ effluxes and high $TCO_2$ production in the jar experiments 7 days after flooding (Fig. 2a & 5). Rapid microbial $CO_2$

production has previously been observed in experiments with experimentally flooded soils (Chambers et al., 2011; Neubauer et al., 2013; Weston et al., 2011). In both soil types, $TCO_2$ production in the surface soil increased over the first 2 months, peaked, and then decreased gradually towards the end. These temporal dynamics were out of phase with DOC availability, indicating that terminally oxidizing microbes may adapt slower to flooded conditions than hydrolyzing microbes. Similar cases of initial substrate hydrolysis outpacing fermentation and SR has been observed before (Arnosti et al., 1994), maybe

due to lag response in the microbial community (Bruchert and Arnosti, 2003). Nevertheless, the majority (~80 %; Table 2) of produced DOC over the whole experiment was oxidized completely to $TCO_2$, while the rest effluxed to the overlying water (~19 %) or accumulated in porewater (~1 %).

### 4.2 OC degradation pathways

$SO_4^{2-}$ was an important electron acceptor in both soils and SR accounted for 63 and 36 % of $TCO_2$ production over the whole

experiment in UC and C soil, respectively (Table 3). One week after flooding, active SR corresponding to 30-40 % of total anaerobic $TCO_2$ production, was detected in the jar experiment. The importance of SR increased gradually over the experiment and by the end accounted for up to 100 % of $TCO_2$ production in both soil types. This is in accordance with Weston et al. (2006) who measured SR in freshwater marsh soil exposed to seawater in anoxic flow through reactors, and found that the relative importance of SR for total $TCO_2$ production increased from 18 % initially to >95 % after 4 weeks.

The delay in SR probably reflects a lag phase for the community of $SO_4^{2-}$ reducing microbes to respond to elevated $SO_4^{2-}$ levels. The delay in SR could also reflect initial competition with other $TCO_2$ producing pathways (e.g. $NO_3^-$ and Fe reduction) in the time right after flooding when $NO_3^-$ and oxidized Fe might have been abundant. However, as the soil became reduced due to increased OC degradation activity and limited $O_2$ supply, electron acceptors other than $SO_4^{2-}$ were rapidly depleted and SR became the dominant respiration pathway.

By combining results from flux and jar experiments it was possible to confine the relative importance of different microbial respiration pathways in flooded soils. The difference between $TCO_2$ effluxes (aerobic + anaerobic processes) and $TCO_2$ production in jar experiments (anaerobic processes) suggested that aerobic respiration only played a minor role in the flooded soils (18 and 10 % in UC and C soil, respectively). On the other hand, SR was quantitatively the most important



pathway, constituting 63 and 36 % of total C-mineralization to $TCO_2$ in UC and C soil, respectively. Hence 19 (UC) to 54 % (C) of $CO_2$ production occurred by respiration processes not directly accounted for. Weston et al. (2006) found that Fe reduction was responsible for about 60 % of $CO_2$ production in the first 4 days after saltwater intrusion in coastal soils. When considering the high initial concentrations and the rapid decrease in soil RFe(III) in our experiment (Fig. 6), respiratory Fe-

reduction was probably an important respiration process initially. However, based on this experiment it was not possible to distinguish between biological and chemical Fe-reduction.

**4.3 Will newly flooded coastal habitats be hotspots for OC burial?**

In this study we observed that only 6-7 % of the total OC pool in coastal soils was degraded by microbial processes in the first year after flooding with seawater. The low final OC degradation rates (especially the very low DOC production)

suggested that remaining OC would be permanently buried due to the limited ability of anaerobic microbial communities to degrade complex organic matter of terrestrial origin (Burdige, 2007; Canfield, 1994; Hedges and Keil, 1995). This means that coastal soils flooded with seawater due to either sea level rise or mitigation techniques such as coastal realignment, will represent a significant C-sink in global C-budgets. For instance, a detailed investigation of the soil characteristics in the topsoils (down to 20 cm) at Gyldensteen Strand suggests $48\pm6\cdot10^3$ kg OC $ha^{-1}$ (average ± SEM, n = 30, T. Valdemarsen,

unpublished results). Hence, when assuming that about 10 % of the soil organic matter will be degraded after flooding, the nature restoration project at Gyldensteen Strand (211 ha) constitutes an immediate C-sink of about $9\cdot10^6$ kg OC (or about 0.02 % of the total annual $CO_2$ emission of Denmark).

**4.4 Efficient Fe-driven sulfide buffering in flooded soils**

Accumulation of free $H_2S$ is often seen in metabolically active organic enriched marine sediments, where it has toxic effects

on benthic fauna (Hargrave et al., 2008; Valdemarsen et al., 2010). It was therefore a concern if free $H_2S$ would accumulate in the soils from Gyldensteen after flooding, since this could hamper the succession of benthic fauna as well as overall ecological developments. However, despite the extremely high initial SR rates in the flooded soils, comparable to SR measured beneath fish farms (Bannister et al., 2014; Holmer et al., 2003) no accumulation of free $H_2S$ occurred in any of the soil types. Weston et al. (2011) also observed a similar lack of $H_2S$ accumulation in soils flooded with seawater, suggesting

that newly flooded soils have a high capacity to buffer $H_2S$. Budget considerations suggest that most of the produced $H_2S$ was immediately re-oxidized, e.g. with $O_2$ in the surface soils, while a significant proportion (33 and 62 % in UC and C soil, respectively) precipitated as different Fe-S compounds, for instance FeS and $Fe_3S_4$ in AVS and $FeS_2$ and $S^0$ in CRS (Rickard and Morse, 2005; Valdemarsen et al., 2010). The depth profiles of solid Fe and S showed that sulfide precipitation occurred at the same depths where active SR was measured, i.e. in the upper 10 cm in UC soil and down to 20 cm depth in C soil.



## 5 Conclusions

In this study a rapid stimulation of heterotrophic microbial degradation of OC was observed in two different soils (uncultivated or cultivated) following flooding with seawater. Degradation rates peaked in the first 2 months after flooding, and hereafter gradually declined to low levels after 1 year. Microbial SR was rapidly established in both soil types and was

the dominating respiration pathway. Nevertheless, despite extremely high SR rates, $H_2S$ did not accumulate in the soils as it was re-oxidized with $O_2$ at soil-water interphase or precipitated with Fe to form AVS and CRS. All three hypotheses stated initially were confirmed. Total OC degradation activity in the tested soils clearly did depend on soil OC content (hypothesis 1) and was 3-fold higher in organic rich uncultivated soil compared to the organic poor cultivated soil. However, only a small proportion of soil OC (6-7 %) was degraded in the first year after flooding, and when considering the low final OC

degradation rates, it appears that most soil OC is non-degradable under anoxic marine conditions and will be preserved after flooding (hypothesis 2). Hence this study suggest that in soils flooded with seawater the majority of soil OC will be permanently preserved creating a negative feedback on atmospheric $CO_2$ concentrations (hypothesis 3).

*Acknowledgements.* We thank technician Birthe Christiansen for help with chemical analyses. Further we thank Erik

Kristensen and Marianne Holmer for valuable discussions and for initiating research at Gyldensteen Strand. This work was funded by the Aage V. Jensen Nature Foundation.

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





**Table 1 Mean values of water content, porosity and organic carbon (OC) for all core sectionings. Error indicated as SEM (n = 15).**

|  | Depth (cm) | Water content (%) | | | Porosity | | | OC (%) | | |
|---|---|---|---|---|---|---|---|---|---|---|
| **Station UC** | 0.5 | 82.9 | ± | 0.7 | 0.82 | ± | 0.04 | 16.2 | ± | 0.8 |
|  | 1.5 | 75.5 | ± | 1.6 | 0.97 | ± | 0.02 | 16.1 | ± | 1.2 |
|  | 3.5 | 60.5 | ± | 1.8 | 0.79 | ± | 0.01 | 11.0 | ± | 0.8 |
|  | 7.5 | 39.3 | ± | 0.9 | 0.60 | ± | 0.01 | 5.2 | ± | 0.2 |
|  | 12.5 | 33.0 | ± | 0.7 | 0.54 | ± | 0.01 | 3.5 | ± | 0.2 |
|  | 17.5 | 34.5 | ± | 0.8 | 0.56 | ± | 0.01 | 3.5 | ± | 0.2 |
| **Station C** | 0.5 | 32.0 | ± | 0.6 | 0.58 | ± | 0.02 | 1.4 | ± | 0.0 |
|  | 1.5 | 24.8 | ± | 0.5 | 0.53 | ± | 0.01 | 1.1 | ± | 0.0 |
|  | 3.5 | 21.6 | ± | 0.3 | 0.40 | ± | 0.01 | 1.0 | ± | 0.0 |
|  | 7.5 | 18.9 | ± | 0.4 | 0.35 | ± | 0.01 | 0.8 | ± | 0.1 |
|  | 12.5 | 17.9 | ± | 0.3 | 0.34 | ± | 0.00 | 0.9 | ± | 0.0 |
|  | 17.5 | 19.8 | ± | 0.4 | 0.37 | ± | 0.01 | 1.0 | ± | 0.0 |





**Table 2** Carbon budget table showing mean organic carbon (OC) ± SEP (n = 18) in uncultivated (UC) and cultivated (C) soil. Total time integrated efflux and accumulation of total carbon dioxide (TCO$_2$) and dissolved organic carbon (DOC) in porewater are also shown.

| Carbon budget (mol m$^{-2}$) | Station UC | Station C |
|---|---|---|
| OC pool | 710.9 ± 54 | 232.5 ± 22 |
| TCO$_2$ efflux | 39.9 | 11.2 |
| DOC efflux | 8.9 | 2.4 |
| TCO$_2$ porewater accumulation | 0.1 | 0.8 |
| DOC porewater accumulation | 0.7 | 0.5 |
| Total OC degradation | 49.6 | 14.8 |
| Percentage of OC pool degraded | 7 % | 6 % |





**Table 3 Budget table based on time integrated total carbon dioxide (TCO$_2$) efflux, and TCO$_2$ production and sulfate reduction (SR) measured in jar experiments (anaerobic rates). Total sulfur precipitated as chromium reducible sulfides (CRS) and acid volatile sulfides (AVS) is also shown.**

| Budget of total degradation to TCO$_2$ (mol m$^{-2}$) | Station UC | Station C |
|---|---|---|
| Total degradation to TCO$_2$ | 40.0 | 12.0 |
| Anaerobic degradation to TCO$_2$ | 32.6 | 10.8 |
| TCO$_2$ production by SR | 25.3 | 4.3 |
| Relative contribution of aerobic respiration | 18 % | 10 % |
| Relative contribution of SR | 63 % | 36 % |
| Relative contribution of other anaerobic respiration | 19 % | 54 % |
| Fe precipitated sulfide accumulation (AVS and CRS) | 4.2 | 1.3 |
| Re-oxidized sulfide | 8.5 | 0.8 |





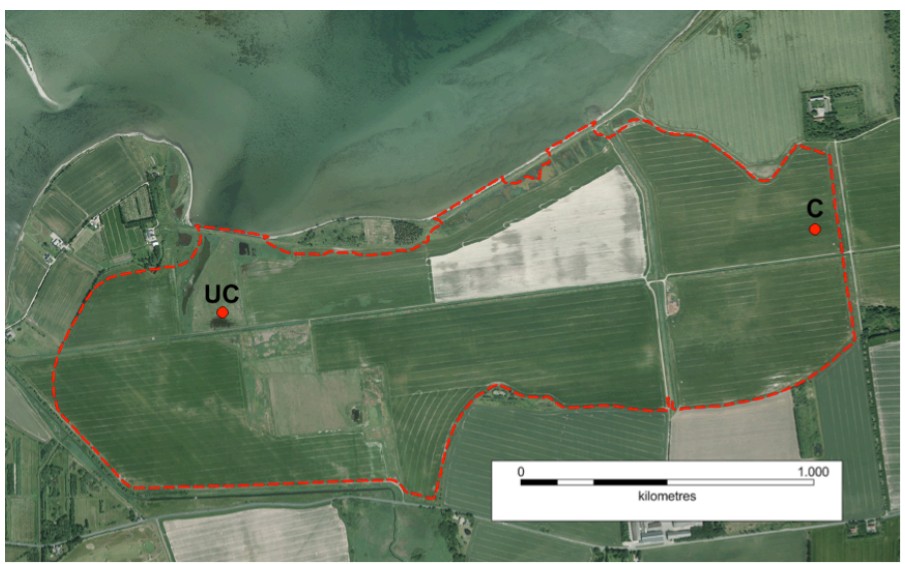

**Figure 1 Map of Gyldensteen Strand with the location of the 2 sampling stations for collecting uncultivated (UC) and cultivated (C) soil cores. The dashed red line indicates the area flooded with seawater in March 2014.**





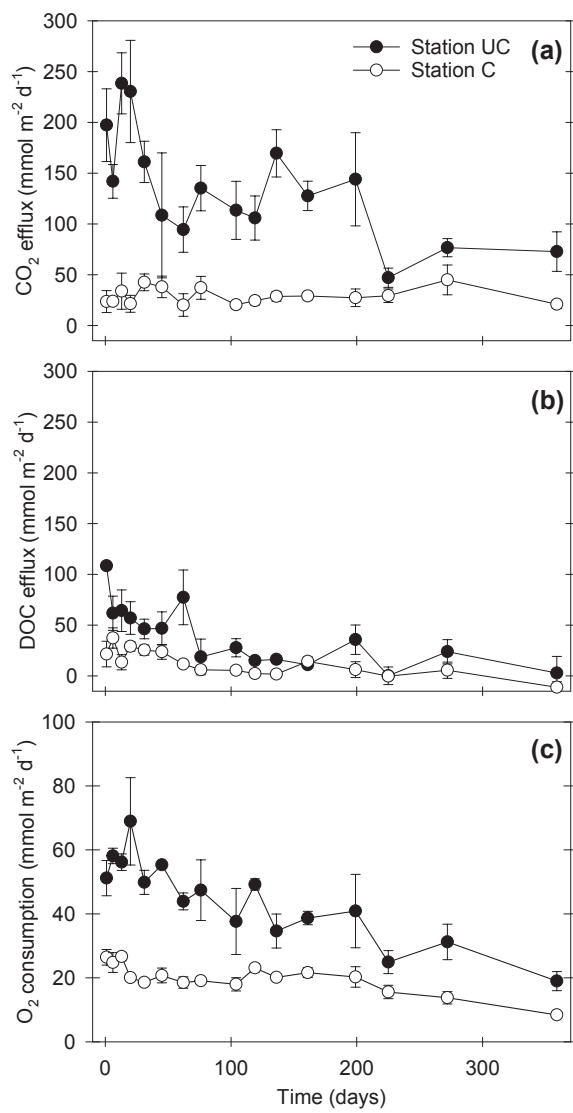

**Figure 2 Fluxes of total carbon dioxide (TCO₂, A), dissolved organic carbon (DOC, B) and oxygen (O₂) consumption (C) in soil cores with uncultivated (UC) and cultivated (C) soil after flooding. Error bars indicate SEM (n = 3).**



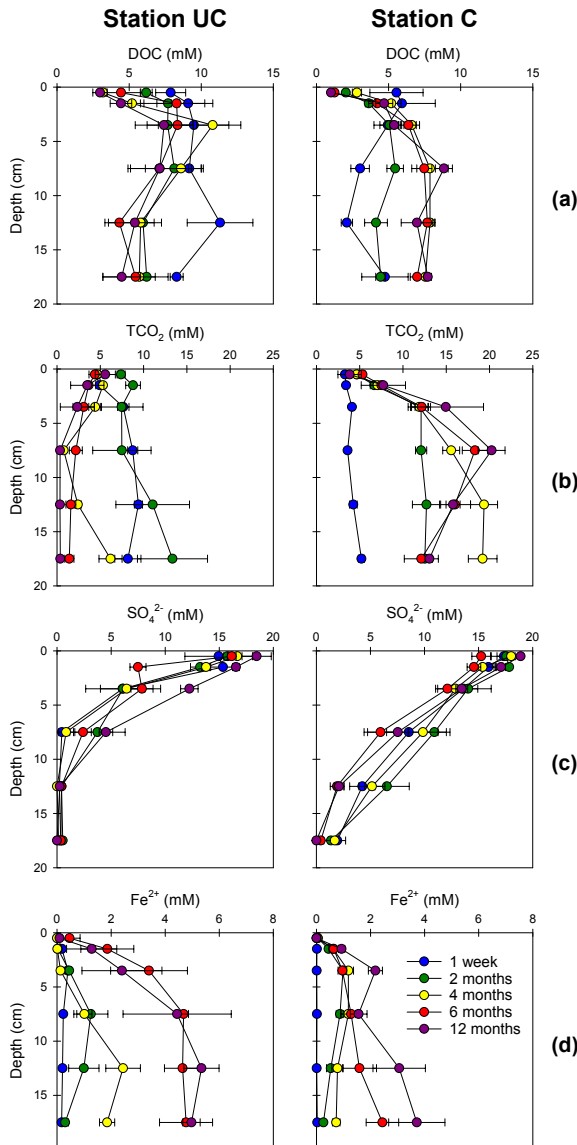

Figure 3 Porewater profiles for dissolved organic carbon (DOC, A), total carbon dioxide (TCO$_2$, B), sulfate (SO$_4^{2-}$) (C) and Fe$^{2+}$ (D) in uncultivated (UC) and cultivated (C) soil flooded with seawater. Error bars indicate SEM (n = 3).





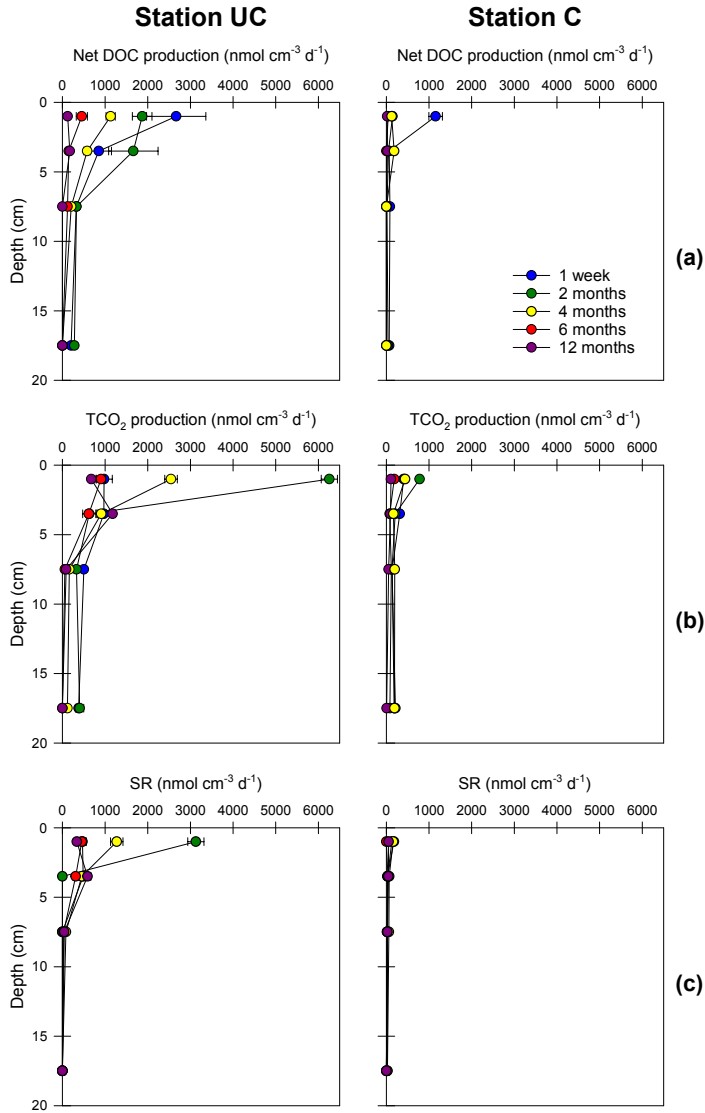

**Figure 4** Temporal and spatial variability in production of dissolved organic carbon (DOC, A) and carbon dioxide ($TCO_2$, B) and sulfate reduction (SR) measured in jar experiments with uncultivated (UC) and cultivated (C) soils flooded with seawater. Error bars indicate SEM.





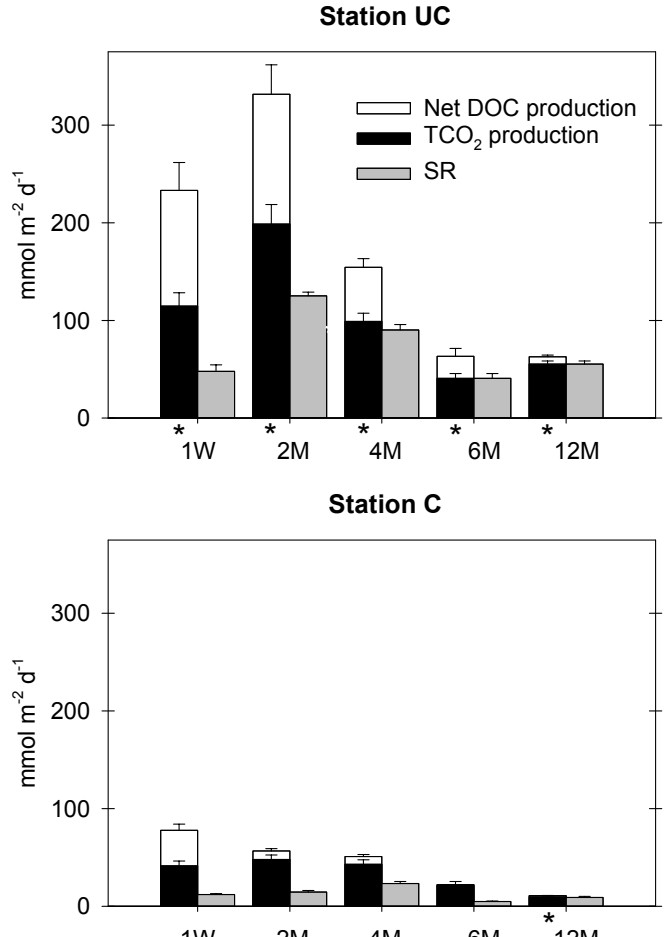

**Figure 5 Results from jar experiments showing area specific net production of dissolved organic carbon (DOC) and total carbon dioxide (TCO₂), and sulfate reduction (SR, based on SR rate measurements converted to C-units) in uncultivated (UC) and cultivated (C) soil at different times after flooding (1 week [1W] and 2, 4, 6 and 12 months [2M, 4M, 6M and 12M, respectively). In columns marked with \*, TCO₂ production was corrected with 2 x SR. Error bars indicate SEP (n = 4).**





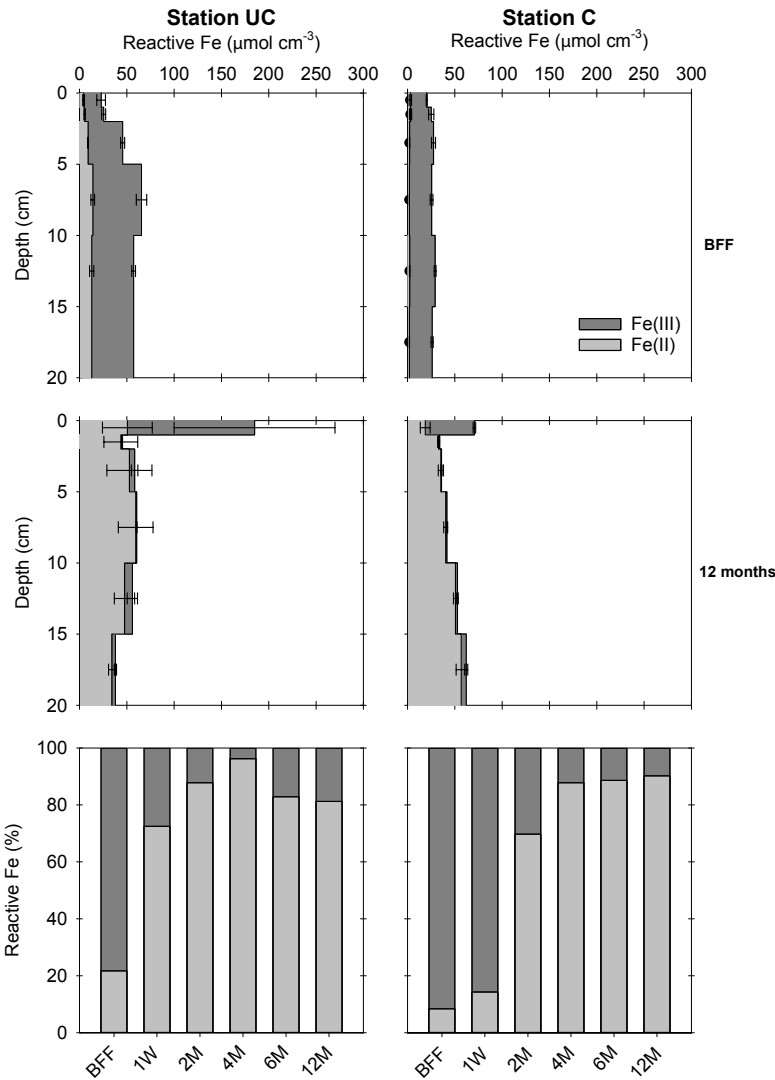

**Figure 6 Upper panels show concentration of reactive Fe(II) and Fe(III) in uncultivated (UC) and cultivated (C) soils before flooding (BFF) and 12 months after flooding. Lower panels show the relative contributions of reactive Fe(II) and Fe(III) in the upper 20 cm at various times after flooding (1 week [1W] and 2, 4, 6 and 12 months [2M, 4M, 6M and 12M], respectively). Error bars indicate SEM (n = 3).**





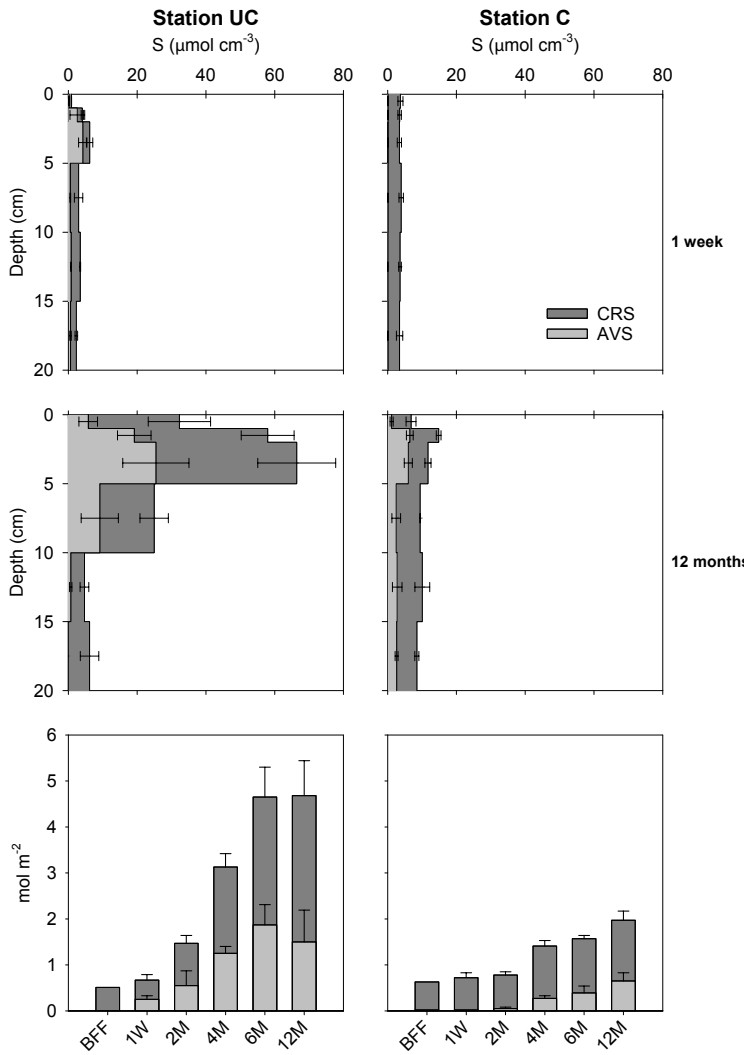

**Figure 7 Upper panels show concentration of chromium reducible sulfides (CRS) and acid volatile sulfides (AVS) in uncultivated (UC) and cultivated (C) soils before flooding (BFF) and 12 months after flooding. Lower panels show the depth integrated pools of AVS and CRS in the upper 20 cm at various times after flooding (1 week [1W] and 2, 4, 6 and 12 months [2M, 4M, 6M and 12M], respectively). Error bars indicate SEM (n = 3).**