# Peer review of "Carbon degradation in agricultural soils flooded with seawater after managed coastal realignment"

_Biogeosciences, 2016_

## Referee Comment (RC1) · Anonymous Referee #1 · 20 Jan 2017

General comments:

authors: Sjogaard et al. This manuscript evaluates the effect of flooding soils with seawater on the carbon mineralisation pathways and rates in these soils. This is clearly a relevant topic with respected to planned managed coastal realignement projects to improve coastal defences against sea level rise. The experiment tackles an environmental issue and seems to be well designed and executed. The manuscript is well written and to the point. However, the major hypothesis (hypothesis 3: does the flooding of soils promote organic carbon preservation?), which is the core and carries the impact of this paper is not well supported (see section below). Furthermore, there are a few more issues and some technical corrections that need revision before this

manuscript is ready to be accepted in BGS. These issues need to be addressed before the manuscript is ready for publication. I recommend major revisions.

Major specific comments:

- Paragraph 4.3: This paragraph is, according to me, the most important conclusion of this manuscript. If coastal soils are re-exposed to marine conditions, will they promote carbon burial and this form a negative feedback on atmospheric CO2 concentrations? Unfortunately, this is also the least documented paragraph, and it does not provide enough evidence to valid such a strong conclusion as posed on P13L11-12 (this study suggests that the majority of soil OC will be permanently preserved . . .).

There is not data or values of pre-flooding mineralisation rates, nor a comparison to normal marine conditions. Furthermore, the TCO2 flux of 67 mmol m-2 d-1 in the uncultivated soil measured by the end of the experiment (and the value of 239 mmol m-2 d-1 on day 13) (see section 3.2) are indications for an extremely high mineralisation rate. The effluxes in the cultivated soil (29 mmol m-2 d-1) indicate normal rates for marine sediments. It is highly likely these rates are transient, and are driven by the DOC production, but this would mean that the standard soil conditions do not produce this DOC, and thus that reinstating marine conditions actually inhibits carbon burial.

Hence, there seems to be no direct evidence that newly flooded coastal habitats will be hotspots for carbon burial. I propose that the authors give a stronger foundation for this paragraph, and show that re-exposure to marine conditions actually decreases the carbon mineralisation (e.g. by providing an estimate of pre-flooding mineralisation rates, or by comparing the carbon burial to pre-flooding carbon burial, and normal marine carbon burial rates).

- Paragraph 4.4: FeIII is indeed efficient as being a sulphide buffer in flooded soils, however, figure 6 shows that virtually all FeIII is converted to FeII by the end of the experiment. This indicates that the FeIII sulphide-buffer was exhausted and sulphide will start accumulating after ∼1 year. This should be mentioned in this paragraph, and

I would also reconsider the term 'efficient buffer' when this would only be active for the time span of 1 year.

- Paragraph 2.3: I have a few remarks/questions for the data analysis that was applied:

When calculating the slopes of the rates in the jar experiments, did you apply any outlier check?

I don't agree with the linear data interpolation that you used to correct for missing data points. In my experience, reactions rates tend to follow exponential trends rather than linear ones. If you want to use this linear interpolation, I would advise to include a small section on the possible errors you make while doing this interpolation.

The correlation you used to convert organic matter to OC units is based on only two points? How did you estimate the significance? Can you show a plot that shows the OM vs the OC, and what model you used?

Minor specific comments:

- P1L11, P3L18: I seem to get a bit confused with the sentence structure. Was station C not in the area that was reflooded? And is the sampling area reflooded, planned to be reflooded or not planned to be reflooded?

- P7L14-21: you mention that the experimental period was not long enough to achieve full saturation of SO4 at 20 cm depth. However, in the C cores, sulphate reaches that depth after the first week, and the concentration at depth decreases over time. This shows that sulphate consumption increases over time (most likely when the FeIII inventory decreases). The UC cores show an increase of sulphate over time and have indeed not achieved saturation at depth.

- P8L11-14: You say that TCO2 production could not be determined below 5 cm depth. You then estimate this TCO2 production by assuming that SR was the dominating pathway at depth. However, when I look at Table 3, you show that the contribution of other anaerobic pathways was 19% for UC and 54% in C, so SR was clearly not the dominant pathway. Also, considering the high FeIII concentrations in the sediment, I would assume that dissimilatory iron reduction is also an important pathway. Considering this, I have some problems with Figure 5, where you show that all TCO2 production from 4 months onwards is due to SR. This is a consequence of your assumption, and I don't feel that this is well founded. Can you provide more justification for this?

- P11L30: I think you can make an estimate of the time evolution of the relative importance of the mineralisation pathways, which could provide more information than the integrated budget over 1 year (since SR will always end up being the dominant pathway if you wait long enough). It would also improve the impact of the manuscript.

- P12L5: Based on the results from the FeIII – measurement of Lovely and Philips, I believe you can estimate the importance of dissimilatory iron reduction (at least, that is what they teach at the AMME summerschool in Odense every year).

Technical corrections:

- Abstract: I find the paper well written, but I don't feel the same about the abstract, it does not flow very well (e.g. 'So far' at the beginning of a sentence). I would advise revising the abstract in order to improve attraction. - P2L18: "Further it is" -> Furthermore it is - P3L25: is the water in the tanks from the same site? If so, please indicate. - P10L9: aerobic OC degradation contributed to 18 and 10 % to of the total . . . - P11L21 anaerobic TCO2 production, was detected -> remove the comma - Figure 3: I would use different symbols for the different months (when printed in black and white, the colors will be difficult to distinguish). - Figure 4: same remark as for figure 3, and I would consider changing the axes of the right panels (it is impossible to see the different SR rates).

---

## Referee Comment (RC2) · Anonymous Referee #2 · 7 Feb 2017

General Comments: Sjøgaard et al. investigate the effect of seawater flooding on the metabolism of soil organic carbon in soils from a reclaimed wetland that had either been cultivated for 140 years or allowed to become a reedswamp. This is an important issue globally as coastal land managers turn to a program of "depoldering" to restore the functions and ecosystem services of intertidal coastal habitats. The author's use bottle and core incubations to explore patterns of carbon mineralization both through time and with soil depth. They conclude that seawater sulfates rapidly accelerate carbon degradation upon flooding, but soils quickly regain a new equilibrium as mineralization slows over time, resulting in only 6-7% of the original soil organic carbon being lost, which they conclude indicates seawater flooding will result in a negative feedback on

atmospheric CO2 concentrations by preserving C. While the analysis conducted were generally well executed and the results comprehensive, they were not designed to test the hypotheses posed in the introduction, specifically (H1) that soil carbon degradation is related to the lability of organic matter, which is not assessed in the current study, (H2) that flooding preserves organic carbon or (H3) there is a negative feedback with soil flooding and atmospheric CO2 concentrations. The results presented are sufficient to support a comprehensive study of the effects of seawater reintroduction to reclaimed coastal lands, but the hypotheses and conclusions must be significantly re-framed to be acceptable for publication. I recommend major revisions.

Major specific comments:

-The abstract is the first mention of coastal realignment but through the manuscript it is discussed as a relatively novel concept about which little is known. There is an extensive body of literature on "managed realignment" also call dike-breach restoration, or depoldering. While I believe the author's data is amongst the most detailed laboratory study of carbon degradation in this body of literature, making it a unique and important addition, they have not used this literature to their advantage and have neglected some key publications, among them the studies of Portnoy and Giblin (Eco. Apps. 1997 pp1054), recent publications by Ardon et al. (GCB 2013 pp296 and Biogeochemistry 2016 411), for a review of biogeochemical changes due to salinization see Herbert et al (2015 Ecosphere) and for reviews of dike-breach restoration see Burdick & Roman (2012) Tidal Marsh Restoration: A Synthesis of Science and Management (Springer)

- There are two problems with the authors' central argument that flooding soils enhances carbon preservation and therefore has a negative feedback with atmospheric CO2 concentrations.

(1) The authors do not show that flooding soils enhances carbon preservation (over what?). The reader may assume that the authors intend to say that flooding the soils preserves more carbon than would be preserved if the land was not subjected to flooding. In fact, the data they present shows flooding increases carbon mineralization, at least initially. While the authors' supposition is not unfounded (intertidal soils on average accumulate carbon at 5-10x the rate of terrestrial soils e.g. McLeod et al 2012) the authors analyses cannot show this because they did not measure mineralization rates in the cultivated and uncultivated soils in the absence of seawater flooding. At the very least the authors could provide a comparison of published rates for similar marine sediments, similar reedswamp sediments, and similar agricultural soils, however this would only be sufficient to suggest, not confirm, enhanced carbon sequestration.

(2) The authors have confused preserving stored carbon with negative carbon-climate feedbacks. Preventing carbon from entering the atmosphere (i.e. through flooding of soils) at best has a null impact on atmospheric $CO_2$ concentrations. To have a negative feedback on atmospheric $CO_2$ concentrations a system must remove $CO_2$ from the atmosphere which is a process not explored in the current MS. Mackey et al (Nature Climate Change 2013) provide an excellent perspective on this. This may well be the case if intertidal vegetation is established etc. but is not the case in the current study.

- Hypothesis 1 & 2: while the dependence of mineralization on content is investigated, the authors do not make any measurements of lability or the origin of organic matter, thus these are weak points of argument that should not be the focus of the manuscript. Instead, the bulk of the analysis are targeted toward bulk organic matter degradation and the pathways of degradation.

-The strongest way to re-frame the data in hand would be a comparison of the effects of seawater flooding on cultivated versus what seem to be freshwater wetland soils focusing on the rates of carbon loss, the proportion of initial carbon lost, and the pathways of mineralization. There are obvious differences in the two sites that lend themselves to this discussion and the topic is still highly relevant to efforts to re-flood former agricultural land (cultivated) as well as restore artificial freshwater impounds (Portnoy & Giblin, Bouldoc & Afton etc.) or the migration of saltwater into historically freshwater areas (absent of restoration).

-Section 4.1 It is the production of small polymers/monomers small enough for microbial uptake that is considered the rate limiting step for carbon degradation (e.g. the enzymatic latch hypotheses) not the generation of DOC which can be highly recalcitrant. DOC is not equivalent to easily degradable materials.

-The authors do not sufficiently address the caveats of long core/bottle incubations and the various experimental artifacts introduced.

Minor specific comments: Abstract o Ln. 6 delete "continue for centuries and" o Ln. 8: delete "So far"; what kind of soils? o Ln. 9: delete "In this study"

Section 1 o Ln. 20: This paragraph invokes far too many specificities related to SLR. Suggest compressing into a single sentence. This paper is about managed realignment with SLR as one of the justifications, not specificities of SLR or scenarios. o Please revisit this argument. It is the hydrolytic enzymes in conjunction with radicals that can depolymerize refractory compounds. There are also multiple other arguments for accelerated decomposition in aerobic environments, including free energy of alternate terminal electron acceptor pathways and other metabolic constraints. o The last two sentences of the first full paragraph starting on page 2 are confusing. Where is "here"? Do the authors intend to say soil organic matter of terrestrial origin may be difficult for marine organisms? o Soil organic carbon is generally abbreviated as SOC

Section 2.1 o Give details about the reclamation: was the area diked and drained? o Because the authors are using so many acronyms for different carbon compounds, the use of "C" and "UC" for the sites can make for difficult reading. Suggest switch to agricultural field (AF=C) and reedswamp (RS=UC) as they are more descriptive. o Was the reedswamp freshwater?

Section 2.2.3 o Were vials flushed to remove oxygen prior to the incubations?

Section 2.3 o The budget calculation is unclear. Please clearly describe which data sources are utilized for the carbon budget.

Section 4.4 o Reduced iron (FeII) responsible for buffering sulfide accumulation (Reddy and DeLaune 2008) appears to increase through most of the study and show no substantial declines (particularly in station C) over the course of the year, indicating there should be sufficient Fe buffer for sulfide generated over longer time scales (>1 year) since metabolic rates appear to decline over time (See Schoepfer et al. 2014. JGR: Biogeossciences).

Technical corrections:

2.2.1 Give simple details of flow injection analysis (model of instrument). Was Zinc added to prevent H2S interference for CO?

Figure 2b. Re-scale y-axis to fit data.

---

## Referee Comment (RC3) · E. Metzger (Referee) · 10 Feb 2017

Dear Editor and co-Authors,

I was happy to have a chance to read this manuscript that provides interesting data about coastal soils that are about to be flooded by seawater under sea level rise. I intended to give constructive comments and suggestions and apologize if some comments seem too harsh, my stylistic skills are quite limited in English.

Edouard Metzger Associated professor at the University of Angers, France

Overall comments:

The study depicted and discussed in the present manuscript represents considerable experimental and analytical work that deserves to be published. In my opinion, the most interesting feature is the almost total replacement of iron oxides by iron sulphides after one year of incubation of a soil with seawater. Such rapid mineralogical transformation suggests rapid anaerobic mineralisation processes that affect the carbonate system and carbon recycling that should be less efficient since the soil tends to become anoxic due to sulfate reduction. Therefore, author main hypothesis is that it should represent a significant negative feedback on atmospheric CO2 concentrations. There is a major objection to such statement due to the lack of elements of comparison. The authors do not show any mineralisation rate of soils before seawater flooding.

Returning to the mineralogical transformation (i.e. iron oxides turned into iron sulphides), nothing is said about potential release of phosphorus and eutrophication that should also have as important feedback even more important to atmospheric CO2. Samples exist, I would suggest to analyse P and address its remobilisation.

The considerable amount of data should permit authors to make a temporal mass balance in order to better precise the relative importance of anaerobic respiration processes and secondary reactions that limit reduced iron and free sulphide diffusion within the sediment and toward the soil water interface.

A secondary point but still important is a better discussion about HCl extractions and iron speciation. There is an extensive literature about selectivity of phases extracted by HCl, dithionite and ascorbic acid that should be considered here (e.g. Kotska and Luther, GCA, 1994; Hyacinthe et al., GCA, 2006).

Then, a quick discussion about the limitations of ex situ long term incubations that cut down hydrosedimentary processes should be addressed.

Summarizing, I recommend a major revision of the manuscript by stepping down on conclusions about carbon preservation and atmospheric CO2 feedback and by examining in more detail the importance of iron cycling.

Abstract:

Should be rewritten in a new version of the manuscript

Introduction:

L64-68 advection processes induced by tidal currents in such porous environment with higher level of connectivity between pores and burrows is not considered here

L83-88 check on in situ experiments by Yucel and Lebris about lignin degradation

Materials and methods:

L152 were cores sliced, centrifuged and conditioned under nitrogen flux?

L162-163 was chloride analysed as well? As a conservative species, chloride is necessary to evaluate sulfate consumption from sulfate profiles in environments of variable salinity. This could refine SR calculations from bulk incubated sediment.

Results

L265-283 Difficult to use TCO2 data since they are potentially compromised. I always recommend to analyse TCO2 or alkalinity as soon as the sample was extracted from the core slice to avoid such disagreement. At this point those date seem invalid for publication.

L284-291 The authors mention that incubation time is too short to achieve full saturation over the entire core. I would agree with that and this can be quite well predicted using diffusive models. For instance the Einstein equation ($x = (Dt)1/2$, Boudreau, 1996) suggest that in 12 month a molecule of sulfate would diffuse in free water at 20°C of about 12 cm (D= 5 10-6 cm-2.s-1, Krom and Berner, 1980). For one week, diffusion allow sulfate to travel only 2 cm. This would suggest that not only diffusion can explain sulfate data and that during pouring of marine water most of it flowed downward through burrows or gaps in the soil. In order to avoid transport processes and to show sulfate consumption from profiles in variable salinity you could, as mentioned

above, to normalise sulfate by chloride: a decrease of the ratio would indicate sulfate consumption that could be quantified in terms of rate to be compared to SR calculated from anoxic incubations. From IC spectra you should be able to retrieve chloride concentrations at least for low salinity samples. I am afraid samples at the higher salinity have to be diluted and reanalysed...

L327-328 add a reference for the SRx2 conversion. Do you achieve a ratio of 2:1 in measured samples? Plotting both measerements should give a nice line with a slope of 2. What about methane oxidation affecting sulfate consumption (1:1 ratio)?

L405-407 You suggest other processes than sulfate reduction to explain carbon mineralisation. You should consider more carefully iron reduction. Data are there to show how important this process is in your soils. This can flaw your main hypothesis that SR is the main mineralisation process going on in your soils after marine water flooding. In a recent study our team showed in intertidal estuarine mudflats that iron reduction remains a major process among sulfate reduction whatever the salinity due to regular replenishment of iron rich particles from the river (Thibault de Chanvalon et al, JSR, 2016). This points out the fact that such long term incubation experiments have somehow to take into account hydrosedimentary processes that can greatly affect organic matter mineralisation.

Discussion

L448-449 Have a look into in situ experiments of wood degradation in marine waters realised by Nadine Lebris team in the Mediterranean (e.g. Yucel et al, Chemosphere, 2013).

L470 The athors claim that SR accounted up to 100% of TCO2 production. How they explain dissolved iron profiles that still show an effective source of reduced iron that should account for a significant part of anaerobic mineralisation processes.

478-481 there is a bias in the statement since it is made from anaerobic incubations.

Data show tha important sulfate reduction occurs near the surface, station UC and is about to be near zero in the other station. What about sulfate reduction if oxygen still diffuses from the surface? You could assess such question looking at porewater profiles. I would be glad to see TCO2 time series and how they fit to linear regressions. Maybe there you can find clues about the relative importance of other mineralisation processes than SR.

L494-495 this final statement underlines the importance of having robust co2 consumption rates: if the sum of iron and sulfate reduction does not achieve mass balance, it becomes to consider other reducing processes for iron in a way and other mineralisation processes in the other

So far, my concerns seem to suggest that a tentative of achievement of mass balnce calculation for C, S and Fe could greatly help interpretations. This could be possible from solid phase speciation and dissolved iron, CO2 and sulfate profiles

502-504 not a sink, at most a zero source. The whole paragraph lacks of evidence. Especially that there are no unflooded cores as reference.

Section 4.4. could be developed by discussing in more detail the switch from FeIII to FeII of the solid phase. It would benefit of mass balance calculations as well. Is there any chance of adding some mineral images or analyses? It would be interesting to look at the crystallinity of iron sulphide minerals formed during the experiment. In marine sediment fromboidal pyrite is formed. I wonder what would be the impact of refractory organic matter on pyrite formation.

Conclusion

Conclusions should be re-drawn according to discussion's evolution.

Figures

Figures 3 and 4: I would suggest a change in colours for different profiles overtime. The grey scale print is very difficult to read. It would be perfect if colours and grey

scale evolve progressively with time and with more contrast.

---

## Author Comment (AC1) · 13 Mar 2017

Dear Dr Slomp, dear reviewers,

we would like to thank all reviewers for taking the time and effort to provide us with such detailed feedback on our manuscript 'Carbon degradation in agricultural soils flooded with seawater after managed coastal realignment'. We have carefully considered all comments and our responses and suggestions on how we will address these can be found below each individual reviewer's comments. Implementing these changes based on the comments of the reviewers will greatly improve our manuscript. We would like to draw attention to the refinements of the argumentation for organic carbon preservation in soils flooded with seawater, especially by incorporating comparisons to relevant

carbon degradation values described in existing literature. Furthermore, the reviewer comments have led us to clarify our description of the experimental conditions and methodology used in the study. We feel that our suggested revisions will improve our manuscript beyond the level necessary to be considered for publication in Biogeosciences.

Kind regards, Kamilla Sjøgaard, Alexander Treusch and Thomas Valdemarsen

Author response comments
General comments: authors: Sjogaard et al. This manuscript evaluates the effect of flooding soils with seawater on the carbon mineralisation pathways and rates in these soils. This is clearly a relevant topic with respected to planned managed coastal re-alignement projects to improve coastal defences against sea level rise. The experiment tackles an environmental issue and seems to be well designed and executed. The manuscript is well written and to the point. However, the major hypothesis (hypothesis 3: does the flooding of soils promote organic carbon preservation?), which is the core and carries the impact of this paper is not well supported (see section below). Furthermore, there are a few more issues and some technical corrections that need revision before this manuscript is ready to be accepted in BGS. These issues need to be addressed before the manuscript is ready for publication. I recommend major revisions.

Major specific comments:

- Paragraph 4.3: This paragraph is, according to me, the most important conclusion of this manuscript. If coastal soils are re-exposed to marine conditions, will they promote carbon burial and this form a negative feedback on atmospheric CO2 concentrations?

Unfortunately, this is also the least documented paragraph, and it does not provide enough evidence to valid such a strong conclusion as posed on P13L11-12 (this study suggests that the majority of soil OC will be permanently preserved . . .).

There is not data or values of pre-flooding mineralisation rates, nor a comparison to normal marine conditions. Furthermore, the TCO2 flux of 67 mmol m-2 d-1 in the uncultivated soil measured by the end of the experiment (and the value of 239 mmol m-2 d-1 on day 13) (see section 3.2) are indications for an extremely high mineralization rate. The effluxes in the cultivated soil (29 mmol m-2 d-1) indicate normal rates for marine sediments. It is highly likely these rates are transient, and are driven by the DOC production, but this would mean that the standard soil conditions do not produce this DOC, and thus that reinstating marine conditions actually inhibits carbon burial. Author response: We acknowledge that pre-flooding mineralization rates would have been good to have, but we did not have the resources to conduct such measurements. In the revised version of the manuscript we will strengthen this part of the manuscript by comparing measured mineralization rates to values available from the literature for comparable marine sediments and agricultural soils. We will add a table to make this argument clear. TCO2 fluxes from Danish coastal marine sediments for comparison are described in Valdemarsen et al. (2010) and Valdemarsen et al. (2014) investigated mineralization rates from a Danish fjord using the same methods as in this experiment. Danish agricultural soils had CO2 effluxes between 42 and 167 mmol m-2 d-1 (Chirinda et al. 2014), which is higher than the fluxes from our experiment at steady state. We see a major acceleration in mineralization at UC, however this is only a short-term effect of leaching and degradation of the labile carbon constituting only a minor fraction of the TOC pool (6-7%). Our experiment indicates that the vast majority of the TOC pool will be preserved long-term after flooding with seawater. Furthermore, we believe that the final CO2 efflux rates for UC are influenced by porewater CO2 diffusing out of the sediment – porewater CO2 that has accumulated in the initial phases of the experiment when mineralization rates were high. When considering the mineralization rates measured in the anoxic incubations at station UC, the final rates are lower than

the TCO2 fluxes (40-55 mmol m-2 d-1).

Hence, there seems to be no direct evidence that newly flooded coastal habitats will be hotspots for carbon burial. I propose that the authors give a stronger foundation for this paragraph, and show that re-exposure to marine conditions actually decreases the carbon mineralisation (e.g. by providing an estimate of pre-flooding mineralization rates, or by comparing the carbon burial to pre-flooding carbon burial, and normal marine carbon burial rates). Author response: Author response: For reasons mentioned above we disagree with the statement that our study does not provide direct evidence for the fact "that flooded soils will become hotspots for OC burial". In the revised manuscript we will strengthen our argumentation for this matter by including more comparisons to the literature regarding mineralization rates in soils and marine sediments and temporal degradation patterns.

- Paragraph 4.4: FeIII is indeed efficient as being a sulphide buffer in flooded soils, however, figure 6 shows that virtually all FeIII is converted to FeII by the end of the experiment. This indicates that the FeIII sulphide-buffer was exhausted and sulphide will start accumulating after ∼1 year. This should be mentioned in this paragraph, and I would also reconsider the term 'efficient buffer' when this would only be active for the time span of 1 year. Author response: We believe that Fe will continue buffering the sulphide production, as sulphide reacts with both oxidized and reduced forms of Fe. At the end of the experiment enough FeII was left to buffer sulphide beyond 1 year. Reviewer #2 also agreed with this (Section 4.4). We will amend the manuscript with the appropriate references, e.g. Rickard and Morse (2005) and the ones suggested by Reviewer #2, to clarify this in the relevant paragraphs of the revised manuscript.

- Paragraph 2.3: I have a few remarks/questions for the data analysis that was applied: When calculating the slopes of the rates in the jar experiments, did you apply any outlier check? Author response: Yes, we conducted a check for obvious outliers.

I don't agree with the linear data interpolation that you used to correct for missing data

points. In my experience, reactions rates tend to follow exponential trends rather than linear ones. If you want to use this linear interpolation, I would advise to include a small section on the possible errors you make while doing this interpolation. Author response: The linear data interpolation was used to estimate mineralization rates at 10-15 cm depth, based on measured mineralization rates at depths above and below. It is true that mineralization rates tend to decrease exponentially from the surface and downwards – the exponential pattern is typically very evident from the surface to a few cm depth, while the variation in mineralization rates with depth appears almost linear below. We therefore think that in this case linear interpolation in between two measured data points is a reasonable way to estimate missing data. The error of using linear contra exponential interpolation between data points will be minor since mineralization rates decrease almost linearly at the relevant depths.

The correlation you used to convert organic matter to OC units is based on only two points? How did you estimate the significance? Can you show a plot that shows the OM vs the OC, and what model you used? Author response: The conversion equation to convert organic matter (OM) into OC was based on 20 data points, that showed a highly significant linear OM – OC relationship in the soils ["OC(%) = 0.0649xOM(%) + 0.0936", $r2=0.9824$, $n=20$]

Minor specific comments:

- P1L11, P3L18: I seem to get a bit confused with the sentence structure. Was station C not in the area that was reflooded? And is the sampling area reflooded, planned to be reflooded or not planned to be reflooded? Author response: Both stations are in the reclaimed area, and also the area that has been flooded in the managed realignment. What we tried to explain is that station C, which is agricultural soil, is representative for the majority of the area, while station UC only represent a minority of the area (reedswamp). This is also visualized in figure 1. As this did not become clear, we will re-phrase the sentence on P3L18 to "Station C however, resembled the majority of the re-flooded area that was farmed since the land reclamation (fertilized, ploughed and

used for monoculture, also illustrated in Fig. 1)"

- P7L14-21: you mention that the experimental period was not long enough to achieve full saturation of SO4 at 20 cm depth. However, in the C cores, sulphate reaches that depth after the first week, and the concentration at depth decreases over time. This shows that sulphate consumption increases over time (most likely when the FeIII inventory decreases). The UC cores show an increase of sulphate over time and have indeed not achieved saturation at depth. Author response: By full saturation we mean the same concentration (or close to) as the overlying water as is typically observed in marine sediments with moderate metabolic activity. In both soil types sulfate concentrations decreased steeply with depth throughout the entire experiment and porewater sulfate in the deeper soils were far from equilibrium with respect to sulfate. We will amend the text to clarify this.

- P8L11-14: You say that TCO2 production could not be determined below 5 cm depth. You then estimate this TCO2 production by assuming that SR was the dominating pathway at depth. However, when I look at Table 3, you show that the contribution of other anaerobic pathways was 19% for UC and 54% in C, so SR was clearly not the dominant pathway. Also, considering the high FeIII concentrations in the sediment, I would assume that dissimilatory iron reduction is also an important pathway. Considering this, I have some problems with Figure 5, where you show that all TCO2 production from 4 months onwards is due to SR. This is a consequence of your assumption, and I don't feel that this is well founded. Can you provide more justification for this? Author response: Regarding table 3, please see the next comment below. In our experiment there was an almost 2:1 relationship between CO2-production and sulfate consumption. This will be mentioned in the revised manuscript. We will also add a reference for the SRx2 conversion (Jørgensen 2006).

- P11L30: I think you can make an estimate of the time evolution of the relative importance of the mineralisation pathways, which could provide more information than the integrated budget over 1 year (since SR will always end up being the dominant

pathway if you wait long enough). It would also improve the impact of the manuscript. Author response: We thank the reviewer for the suggestion and acknowledge that the budget in table 3 doesn't illustrate this point. In the revised manuscript we will change table 3 to indicate the total values, as well as temporal development in contribution of mineralization pathways. This will also contribute to the argumentation for carbon preservation.

- P12L5: Based on the results from the FeIII – measurement of Lovely and Philips, I believe you can estimate the importance of dissimilatory iron reduction (at least, that is what they teach at the AMME summerschool in Odense every year). Author response: It is true that there are some tentative correlations between the FeIII content in marine sediments and relative contribution of Fe-reduction to total OC-mineralization – see fig. 6 in Jensen et al. (2003). But this relationship only holds for marine sediments under steady state conditions – not in this case where we are far from steady state.

Technical corrections:

- Abstract: I find the paper well written, but I don't feel the same about the abstract, it does not flow very well (e.g. 'So far' at the beginning of a sentence). I would advise revising the abstract in order to improve attraction. Author response: We will revise the abstract as suggested by the reviewer.

- P2L18: "Further it is" -> Furthermore it is - P3L25: is the water in the tanks from the same site? If so, please indicate. -P10L9: aerobic OC degradation contributed to 18 and 10 % to of the total . . . - P11L21 anaerobic TCO2 production, was detected -> remove the comma - Figure 3: I would use different symbols for the different months (when printed in black and white, the colors will be difficult to distinguish). - Figure 4: same remark as for figure 3, and I would consider changing the axes of the right panels (it is impossible to see the different SR rates). Author response: We will include the above corrections in the revised manuscript.

Interactive comment on "Carbon degradation in agricultural soils flooded with seawater

after managed coastal realignment" by Kamilla S. Sjøgaard et al. Anonymous Referee #2

General Comments: Sjøgaard et al. investigate the effect of seawater flooding on the metabolism of soil organic carbon in soils from a reclaimed wetland that had either been cultivated for 140 years or allowed to become a reedswamp. This is an important issue globally as coastal land managers turn to a program of "depoldering" to restore the functions and ecosystem services of intertidal coastal habitats. The author's use bottle and core incubations to explore patterns of carbon mineralization both through time and with soil depth. They conclude that seawater sulfates rapidly accelerate carbon degradation upon flooding, but soils quickly regain a new equilibrium as mineralization slows over time, resulting in only 6-7% of the original soil organic carbon being lost, which they conclude indicates seawater flooding will result in a negative feedback on atmospheric $CO_2$ concentrations by preserving C. While the analysis conducted were generally well executed and the results comprehensive, they were not designed to test the hypotheses posed in the introduction, specifically (H1) that soil carbon degradation is related to the lability of organic matter, which is not assessed in the current study, (H2) that flooding preserves organic carbon or (H3) there is a negative feedback with soil flooding and atmospheric $CO_2$ concentrations. The results presented are sufficient to support a comprehensive study of the effects of seawater reintroduction to reclaimed coastal lands, but the hypotheses and conclusions must be significantly re-framed to be acceptable for publication. I recommend major revisions.

Major specific comments:

-The abstract is the first mention of coastal realignment but through the manuscript it is discussed as a relatively novel concept about which little is known. There is an extensive body of literature on "managed realignment" also call dike-breach restoration, or depoldering. While I believe the author's data is amongst the most detailed laboratory study of carbon degradation in this body of literature, making it a unique and important addition, they have not used this literature to their advantage and have neglected
some key publications, among them the studies of Portnoy and Giblin (Eco. Apps. 1997 pp1054), recent publications by Ardon et al. (GCB 2013 pp296 and Biogeochemistry 2016 411), for a review of biogeochemical changes due to salinization see Herbert et al (2015 Ecosphere) and for reviews of dike-breach restoration see Burdick & Roman (2012) Tidal Marsh Restoration: A Synthesis of Science and Management (Springer) Author response: We thank the reviewer for considering our study 'amongst the most detailed laboratory study of carbon degradation in this body of literature'. While it is always difficult to find and incorporate all available knowledge into manuscripts, we will certainly read and incorporate the studies suggested by the reviewer into the revised version of the manuscript.

- There are two problems with the authors' central argument that flooding soils enhances carbon preservation and therefore has a negative feedback with atmospheric CO2 concentrations. (1) The authors do not show that flooding soils enhances carbon preservation (over what?). The reader may assume that the authors intend to say that flooding the soils preserves more carbon than would be preserved if the land was not subjected to flooding. In fact, the data they present shows flooding increases carbon mineralization, at least initially. While the authors' supposition is not unfounded (intertidal soils on average accumulate carbon at 5-10x the rate of terrestrial soils e.g. McLeod et al 2012) the authors analyses cannot show this because they did not measure mineralization rates in the cultivated and uncultivated soils in the absence of seawater flooding. At the very least the authors could provide a comparison of published rates for similar marine sediments, similar reed swamp sediments, and similar agricultural soils, however this would only be sufficient to suggest, not confirm, enhanced carbon sequestration. Author response: As described in the first response comment to reviewer 1#, we will refine the argumentation and provide comparisons to published rates.

(2) The authors have confused preserving stored carbon with negative carbon-climate feedbacks. Preventing carbon from entering the atmosphere (i.e. through flooding

of soils) at best has a null impact on atmospheric CO2 concentrations. To have a negative feedback on atmospheric CO2 concentrations a system must remove CO2 from the atmosphere which is a process not explored in the current MS. Mackey et al (Nature Climate Change 2013) provide an excellent perspective on this. This may well be the case if intertidal vegetation is established etc. but is not the case in the current study. Author response: We have used the term "negative feedback on atmospheric CO2 concentrations" to describe the fact that, due to the preservation of the organic C present in the soils at the time of flooding, on a longer term less CO2 will be emitted than under a scenario where the area would not have been flooded. We believe that this is the correct understanding of the term 'negative feedback'. In the revised version of the manuscript we will consult the manuscript suggested by the reviewer and assess if our argumentation should be refined.

- Hypothesis 1 & 2: while the dependence of mineralization on content is investigated, the authors do not make any measurements of lability or the origin of organic matter, thus these are weak points of argument that should not be the focus of the manuscript. Instead, the bulk of the analysis are targeted toward bulk organic matter degradation and the pathways of degradation. Author response: There's a considerable amount of literature discussing the importance of labile and refractory organic matter from bulk C-degradation rates (Westrich and Berner 1984, Burdige 1991, Valdemarsen et al. 2014). The exponentially decreasing trend in C-degradation can only be explained by a gradual depletion of the most labile components of soil organic C. This will be clarified in the revised manuscript.

-The strongest way to re-frame the data in hand would be a comparison of the effects of seawater flooding on cultivated versus what seem to be freshwater wetland soils focusing on the rates of carbon loss, the proportion of initial carbon lost, and the pathways of mineralization. There are obvious differences in the two sites that lend themselves to this discussion and the topic is still highly relevant to efforts to re-flood former agricultural land (cultivated) as well as restore artificial freshwater impoundments (Portnoy

& Giblin, Bouldoc & Afton etc.) or the migration of saltwater into historically freshwater areas (absent of restoration). Author response: We thank the reviewer for his/her suggestion to rewrite the manuscript with an alternative angle. However, rephrasing the manuscript would be counterproductive in relation to the main motivation for the manuscript, which was to assess the total impact of flooding on soil C degradation in the specific area that was flooded.

-Section 4.1 It is the production of small polymers/monomers small enough for microbial uptake that is considered the rate limiting step for carbon degradation (e.g. the enzymatic latch hypotheses) not the generation of DOC which can be highly recalcitrant. DOC is not equivalent to easily degradable materials. Author response: The reviewer is right that it is 'the production of small polymers/monomers small enough for microbial uptake that is considered the rate limiting step for carbon degradation', but in most cases by far, most of the DOC produced per time unit is 'small polymers/monomers small enough for microbial uptake' while only a small proportion is recalcitrant DOC. However, over time recalcitrant DOC may accumulate to high (and quantitatively important) levels in soil porewater. We will make sure that this point is adequately described in the revised manuscript.

-The authors do not sufficiently address the caveats of long core/bottle incubations and the various experimental artifacts introduced. Author response: We will include a section discussing the potential influences that our experimental setup had on our results, e.g. day/night cycles of light and temperature, daily water exchange due to tides, very undisturbed soils in the experiment due to e.g. lacking of fauna.

Minor specific comments: Abstract o Ln. 6 delete "continue for centuries and" o Ln. 8: delete "So far"; what kind of soils? o Ln. 9: delete "In this study" Author response: We will adjust the abstract to the changes after revision of the manuscript and revise it accordingly.

Section 1 o Ln. 20: This paragraph invokes far too many specificities related to SLR.

Suggest compressing into a single sentence. This paper is about managed realignment with SLR as one of the justifications, not specificities of SLR or scenarios. o Please revisit this argument. It is the hydrolytic enzymes in conjunction with radicals that can depolymerize refractory compounds. There are also multiple other arguments for accelerated decomposition in aerobic environments, including free energy of alternate terminal electron acceptor pathways and other metabolic constraints. o The last two sentences of the first full paragraph starting on page 2 are confusing. Where is "here"? Do the authors intend to say soil organic matter of terrestrial origin may be difficult for marine organisms? o Soil organic carbon is generally abbreviated as SOC Author response: We will reduce the paragraph on SLR (sea level rise) to one or two sentences and fuse it with the following paragraph that introduces managed coastal realignment. All the other minor points mentioned by the reviewer related to this part of the manuscript will also be addressed in the revised manuscript

Section 2.1 o Give details about the reclamation: was the area diked and drained? Author response: This is correct. In the revised manuscript "reclaimed" will be substituted with "diked and continuously drained".

o Because the authors are using so many acronyms for different carbon compounds, the use of "C" and "UC" for the sites can make for difficult reading. Suggest switch to agricultural field (AF=C) and reedswamp (RS=UC) as they are more descriptive. o Was the reedswamp freshwater? Author response: We thank the reviewer for an alternative suggestion for abbreviations, but we prefer to keep our original terminology.

Section 2.2.3 o Were vials flushed to remove oxygen prior to the incubations? Author response: Vials were not flushed prior to incubations.

Section 2.3 o The budget calculation is unclear. Please clearly describe which data sources are utilized for the carbon budget. Author response: We will improve the description of budget calculations.

Section 4.4 o Reduced iron (FeII) responsible for buffering sulfide accumulation (Reddy

and DeLaune 2008) appears to increase through most of the study and show no substantial declines (particularly in station C) over the course of the year, indicating there should be sufficient Fe buffer for sulfide generated over longer time scales (>1 year) since metabolic rates appear to decline over time (See Schoepfer et al. 2014. JGR: Biogeossciences). Author response: In the revised manuscript, the argument related to sulfide buffering will be revised according to the reviewer's suggestion, with e.g. the incorporation of the suggested references.

Technical corrections:

2.2.1 Give simple details of flow injection analysis (model of instrument). Was Zinc added to prevent H2S interference for CO? Author response: The instrument for flow injection analysis is exactly as described in the reference provided (Hall and Aller 1992), so we feel that no additional description is needed. Saturated HgCl2 was added, which also precipitates sulfide through the formation of HgS.

Figure 2b. Re-scale y-axis to fit data. Author response: We will perform the suggested change in the revised manuscript.
Dear Editor and co-Authors, I was happy to have a chance to read this manuscript that provides interesting data about coastal soils that are about to be flooded by seawater under sea level rise. I intended to give constructive comments and suggestions and apologize if some comments seem too harsh, my stylistic skills are quite limited in English. Edouard Metzger Associated professor at the University of Angers, France

Overall comments:

The study depicted and discussed in the present manuscript represents considerable experimental and analytical work that deserves to be published. In my opinion, the

most interesting feature is the almost total replacement of iron oxides by iron sulphides after one year of incubation of a soil with seawater. Such rapid mineralogical transformation suggests rapid anaerobic mineralisation processes that affect the carbonate system and carbon recycling that should be less efficient since the soil tends to become anoxic due to sulfate reduction. Therefore, author main hypothesis is that it should represent a significant negative feedback on atmospheric CO2 concentrations. There is a major objection to such statement due to the lack of elements of comparison. The authors do not show any mineralisation rate of soils before seawater flooding. Author response: Reviewer #1 and #2 have also mentioned that we lack a comparison to mineralization rates in soils before flooding. As stated in the response to reviewer #1's comment, the revised manuscript will include comparisons to typical mineralization rates in soils as well as improved arguments related to this matter.

Returning to the mineralogical transformation (i.e. iron oxides turned into iron sulphides), nothing is said about potential release of phosphorus and eutrophication that should also have as important feedback even more important to atmospheric CO2. Samples exist, I would suggest to analyse P and address its remobilisation. Author response: While we acknowledge that P-release from newly flooded sediments is an extremely important process in relation to eutrophication issues, we have tried to write a focused manuscript dealing with C-, Fe- and S-cycling. Adding P-cycling would make it impossible to keep the story tight and focused, as we would have to discuss many more processes in detail related to P-cycling, as well as broader eutrophication issues. We therefore prefer to not include P at this point.

The considerable amount of data should permit authors to make a temporal mass balance in order to better precise the relative importance of anaerobic respiration processes and secondary reactions that limit reduced iron and free sulphide diffusion within the sediment and toward the soil water interface. Author response: We agree with the comment and temporal mass balances for C, S and Fe are already included in the manuscript – see figures 5, 6 and 7. Furthermore, we already will address a similar comment of reviewer #1 at "- P11L30:" under "Minor specific comments" regarding table 3, which will also clarify the temporal mass balance.

A secondary point but still important is a better discussion about HCl extractions and iron speciation. There is an extensive literature about selectivity of phases extracted by HCl, dithionite and ascorbic acid that should be considered here (e.g. Kotska and Luther, GCA, 1994; Hyacinthe et al., GCA, 2006). Author response: We do not think that a detailed discussion of Fe-extraction methodology is relevant for the story. However, in the revised manuscript we will improve the description of Fe-extractions and include relevant references to illustrate which Fe-pools are extracted by the chosen method.

Then, a quick discussion about the limitations of ex situ long term incubations that cut down hydrosedimentary processes should be addressed. Author response: Reviewer #2 has already recommended this above (the comment after "Section 4.1"), and we will address the potential influences that our experimental setup had on the results in a new paragraph.

Summarizing, I recommend a major revision of the manuscript by stepping down on conclusions about carbon preservation and atmospheric CO2 feedback and by examining in more detail the importance of iron cycling.

Abstract:

Should be rewritten in a new version of the manuscript Author response: The abstract of the revised manuscript will be thoroughly revised.

Introduction:

L64-68 advection processes induced by tidal currents in such porous environment with higher level of connectivity between pores and burrows is not considered here Author response: We do not think including information about tidal currents and soil porosity would add to the overall story and improve the indicated passage.

L83-88 check on in situ experiments by Yucel and Lebris about lignin degradation Author response: Thank you for making us aware of this, we will check the suggested reference.

Materials and methods:

L152 were cores sliced, centrifuged and conditioned under nitrogen flux? Author response: Cores were sliced under normal atmosphere.

L162-163 was chloride analysed as well? As a conservative species, chloride is necessary to evaluate sulfate consumption from sulfate profiles in environments of variable salinity. This could refine SR calculations from bulk incubated sediment. Author response: Chloride was measured as a proxy for the progress of the intrusion of seawater into the core. However, in this highly unusual case, with virtually all dissolved components (including Cl) far from being at steady state, it is, to our knowledge, very difficult, if not impossible, to use chloride data to correct for sulfate consumption.

Results

L265-283 Difficult to use TCO2 data since they are potentially compromised. I always recommend to analyse TCO2 or alkalinity as soon as the sample was extracted from the core slice to avoid such disagreement. At this point those date seem invalid for publication. Author response: As mentioned in the manuscript (P8 L12-16) we only use the CO2 data that are not compromised by experimental artefacts.

L284-291 The authors mention that incubation time is too short to achieve full saturation over the entire core. I would agree with that and this can be quite well predicted using diffusive models. For instance the Einstein equation ($x = (Dt)1/2$, Boudreau, 1996) suggest that in 12 month a molecule of sulfate would diffuse in free water at 20_C of about 12 cm (D= 5 10-6 cm-2.s-1, Krom and Berner, 1980). For one week, diffusion allow sulfate to travel only 2 cm. This would suggest that not only diffusion can explain sulfate data and that during pouring of marine water most of it flowed downward through burrows or gaps in the soil. In order to avoid transport processes and to show sulfate consumption from profiles in variable salinity you could, as mentioned above, to normalise sulfate by chloride: a decrease of the ratio would indicate sulfate consumption that could be quantified in terms of rate to be compared to SR calculated from anoxic incubations. From IC spectra you should be able to retrieve chloride concentrations at least for low salinity samples. I am afraid samples at the higher salinity have to be diluted and reanalysed. . . Author response: True, the initial water infiltration during flooding was also a transport mechanism for sulfate. This information will be added in the revised manuscript. While we agree that it could be interesting to evaluate if sulfate reduction could also be measured by considering chloride/sulfate ratios in porewater, we believe that the suggested method would be subject to large errors as we are dealing with an experimental system far from steady state. We therefore prefer to base our discussion on direct sulfate consumption measurements, which is usually a very precise method to obtain sulfate reduction measurements – see e.g. (Kristensen and Hansen 1995, Valdemarsen et al. 2012)

L327-328 add a reference for the SRx2 conversion. Do you achieve a ratio of 2:1 in measured samples? Plotting both measerements should give a nice line with a slope of 2. What about methane oxidation affecting sulfate consumption (1:1 ratio)? Author response: We will add a reference for the SRx2 conversion (Jørgensen 2006). We observed an almost 2:1 relationship between $CO_2$-production and sulfate consumption in our experiment, indicating that the influence of methane oxidation was negligible. This will be mentioned in the revised manuscript.

L405-407 You suggest other processes than sulfate reduction to explain carbon mineralisation. You should consider more carefully iron reduction. Data are there to show how important this process is in your soils. This can flaw your main hypothesis that SR is the main mineralisation process going on in your soils after marine water flooding. In a recent study our team showed in intertidal estuarine mudflats that iron reduction remains a major process among sulfate reduction whatever the salinity due to regular

replenishment of iron rich particles from the river (Thibault de Chanvalon et al, JSR, 2016). This points out the fact that such long term incubation experiments have somehow to take into account hydrosedimentary processes that can greatly affect organic matter mineralisation. Author response: As mentioned by the reviewer, systems with high importance of Fe-reduction are systems where Fe-oxides are continuously replenished e.g. by either sedimentation of Fe-oxide-rich particles or intense vertical mixing of the sediment matrix due to bioturbation. Fe-reduction may also be of high relative importance in Fe-rich sediments with low metabolic activity. However, in our experimental system where organic matter of high lability is present in excess initially, sulfate reduction will become the dominating pathway. This becomes very clear from our direct measurements of $CO_2$ production and sulfate consumption, showing that sulfate reduction was responsible for 63 and 36% of total organic carbon degradation (P10 L6). We do not see how we can use our results to quantify Fe-reduction, as oxidized Fe may have been reduced due to both bacterial and chemical reduction and we have no method to distinguish between the two.

Discussion

L448-449 Have a look into in situ experiments of wood degradation in marine waters realised by Nadine Lebris team in the Mediterranean (e.g. Yucel et al, Chemosphere, 2013). Author response: Thank you for bringing this to our attention, we will include this reference and compare the SR rates.

L470 The athors claim that SR accounted up to 100% of TCO2 production. How they explain dissolved iron profiles that still show an effective source of reduced iron that should account for a significant part of anaerobic mineralisation processes. Author response: The high importance of SR was evaluated based on direct measurements of sulfate consumption and CO2-production, and we do not see how this method can be questioned. We agree that any proportion of TCO2 production not explained by SR, may have been due to Fe-reduction.

478-481 there is a bias in the statement since it is made from anaerobic incubations. Data show tha important sulfate reduction occurs near the surface, station UC and is about to be near zero in the other station. What about sulfate reduction if oxygen still diffuses from the surface? You could assess such question looking at porewater profiles. I would be glad to see TCO2 time series and how they fit to linear regressions. Maybe there you can find clues about the relative importance of other mineralisation processes than SR. Author response: We are not sure how to understand the reviewer's comment. In our experimental setup the water overlying the cores was oxygenated and oxygen continuously diffused from the overlying water into the soil cores. The reason that sulfate reduction was occurring close to the sediment surface is that all oxygen was consumed in the uppermost soil layers due to extremely high metabolic rates of aerobic microorganisms initially.

L494-495 this final statement underlines the importance of having robust co2 consumption rates: if the sum of iron and sulfate reduction does not achieve mass balance, it becomes to consider other reducing processes for iron in a way and other mineralisation processes in the other

So far, my concerns seem to suggest that a tentative of achievement of mass balnce calculation for C, S and Fe could greatly help interpretations. This could be possible from solid phase speciation and dissolved iron, CO2 and sulfate profiles Author response: No doubt Fe-reduction may have been a metabolic pathway of minor importance in our experiment. However, we cannot estimate the importance of microbial Fe-reduction since Fe may have been reduced by two competing processes – microbial reduction or spontaneous chemical reduction by sulphide – and we do not have a way to estimate the relative importance of the two.

502-504 not a sink, at most a zero source. The whole paragraph lacks of evidence. Especially that there are no unflooded cores as reference. Author response: This has also been mentioned by reviewer #2. We acknowledge this and will address this matter in the revised manuscript.

Section 4.4. could be developed by discussing in more detail the switch from FeIII to FeII of the solid phase. It would benefit of mass balance calculations as well. Is there any chance of adding some mineral images or analyses? It would be interesting to look at the crystallinity of iron sulphide minerals formed during the experiment. In marine sediment fromboidal pyrite is formed. I wonder what would be the impact of refractory organic matter on pyrite formation. Author response: The presented data clearly documents a switch in Fe-speciation from the domination of oxidized Fe initially, to almost exclusively reduced Fe by the end. Unfortunately we do not have mineral images of any kind and will therefore not be able to add information obtained from image analysis in the revised manuscript.

Conclusion Conclusions should be re-drawn according to discussion's evolution. Author response: The conclusions will be re-written after the revisions according to all reviewer comments have been implemented.

Figures Figures 3 and 4: I would suggest a change in colours for different profiles overtime. The grey scale print is very difficult to read. It would be perfect if colours and grey scale evolve progressively with time and with more contrast. Author response: We will implement the suggestion of the reviewer in the updated version of the manuscript.

References

Burdige, D. J. 1991. The kinetics of organic matter mineralization in anoxic marine sediments. Journal of Marine Research 49:727-761. Chirinda, N., L. Elsgaard, I. K. Thomsen, G. Heckrath, and J. E. Olesen. 2014. Carbon dynamics in topsoil and subsoil along a cultivated toposequence. Catena 120:20-28. Hall, P., and R. Aller. 1992. Rapid, Small-Volume, Flow Injection Analysis for 2 and NH4 + in Marine and Freshwaters. Limnology and Oceanography 37:1113-1119. Jensen, M. M., B. Thamdrup, S. Rysgaard, M. Holmer, and H. Fossing. 2003. Rates and regulation of microbial iron reduction in sediments of the Baltic-North Sea transition. Biogeochemistry 65:295–317. Jørgensen, B. B. 2006. Bacteria and Marine Biogeochemistry. Pages 169-201 in H. D.

Schulz and M. Zabel, editors. Marine Geochemistry. Springer, Berlin Heidelberg New York. Kristensen, E., and K. Hansen. 1995. Decay of plant detritus in organic-poor marine sediment: Production rates and stoichiometry of dissolved C and N compounds. Journal of Marine Research 53:675-702. Rickard, D., and J. W. Morse. 2005. Acid volatile sulfide (AVS). Marine Chemistry 97:141-197. Valdemarsen, T., R. J. Bannister, P. K. Hansen, M. Holmer, and A. Ervik. 2012. Biogeochemical malfunctioning in sediments beneath a deep-water fish farm. Environ Pollut 170:15-25. Valdemarsen, T., E. Kristensen, and M. Holmer. 2010. Sulfur, carbon, and nitrogen cycling in faunated marine sediments impacted by repeated organic enrichment. Marine Ecology Progress Series 400:37-53. Valdemarsen, T., C. O. Quintana, E. Kristensen, and M. R. Flindt. 2014. Recovery of organic-enriched sediments through microbial degradation: implications for eutrophic estuaries. Marine Ecology Progress Series 503:41-58. Westrich, J. T., and R. A. Berner. 1984. The role of sedimentary organic matter in bacterial sulfate reduction: The G model tested. Limnol. Oceanogr. 29:236-249.

---

## Referee Report (RR1)

[referee-annotated manuscript omitted]

---

## Author Response (AR2)

**Dear editor,**

We have now carefully addressed the comments from the two reviewers concerning the revised version of the manuscript. Our responses to individual comments are outlined in red in the text below. We are happy to see that both reviewers are generally positive towards the manuscript.

We have given special concern to the critical comments from Reviewer#2 regarding the wider implications of this study. We agree that we may initially have been too bold when describing the implications of our study. In the revised manuscript we have markedly down-played the climate aspects of the story.

We hope that the revised manuscript is acceptable for publication in Biogeosciences!

Sincerely,

Kamilla S. Sjøgaard

**Response to comments from Reviewer#1**

Dear dr. Sjøgaard,

Thank you for this substantial revision of the manuscript, it does look much better now, and I only have a few (minor) comments and suggestions left.

> *Author response: We thank Reviewer#1 for constructive criticism. We hope we have managed to respond to her/his comments and suggestions in a satisfactory manner.*

Reply to comment 1.6:

I was not very clear in this comment, my apologies. My question was: If you analyzed OC of only 2 time points, did that proved a big enough range to warrant a linear regression (i.e., if the OC vs. OM contents cluster at the two end-points of the regression, you will have a significant regression, but the equation will not necessarily be correct as the slope will be very dependent on the endpoints). Could you perhaps add a plot of the OM vs OC as Supplementary Information?

> *Author response: The datapoints were more or less evenly distributed within the range of the dataset and clustering was not a problem. The regression can be seen below. We do not think these data are so important that they should be added as supplementary material, but will do so if needed.*

[Figure]

P4L14: were the cores sliced in an anoxic atmosphere?

> *Author response: No*

P5L17: did you use a statistical test to check for outliers? Or did you remove them on sight (which is acceptable, if they are obvious, but that has to be mentioned)

> *Author response: No, no formal statistical outlier tests were performed. Line was rephrased to: "…changes by linear regressions after removing obvious outliers (visual check)."*

P6L5: 'accumulated porewater TCO2 at different time points …' -> I assume that this is the accumulation, correct for the accumulation at the time before (i.e. integrated PW TCO2 at month 2 - integrated PW TCO2 at week 1 = produced TCO2 between week 1 and month 2)?

*Author response: Correct.*

P6L16-18: Do you have any idea of the accumulation rate of the sediment? Can you estimate which depth part of the soil is still marine? Considering that you can still find shell material, some part will be the old marine sediment. And thus, how much of the organic carbon is actually soil organic carbon?

*Author response: This is very difficult as the area has been used for agriculture and has been mechanically reworked. We will at a later stage attempt to do deep cores in the area to check if we can find a "marine signal" deep in the soil.*

Section 3.2 and lower: give standard deviations if you show average values

*Author response: Done*

Section 4.2: the two paragraphs are kind of repetitive (first one talks about SR and mentions O2 and other electron acceptors, the second one talks about O2 and others and mentions SR); you could shorten this by combining them and streamlining the text.

*Author response: Thanks for the suggestion. We prefer to keep the two paragraphs as they are. We think mixing the information in the two paragraphs will result in confusion.*

Section 4.4: it might be nice to try and make a rough estimation of the time scale over which this buffering will stay active (assuming the rate of sulfate reduction at the end + the percentage that precipitates + the content of reactive non –sulfurized FeII and FeIII left.

*Author response: We think it is very difficult to come up with a credible estimate based on simple considerations. Especially since more Fe is available for precipitation with sulfide than we can account for – total Fe extractions require much harsher extraction methods than utilized here.*

P12L25 could these high effluxes of CO2 also be due to the absence of CaCO3 precipitation, or is this a negligible effect?

*Author response: This part was revised in response to a comment by Reviewer#2 and comment is no longer relevant.*

Table 3: the relative contribution numbers do not seem to add up, so I am not sure if I understand the table correctly. First row:

    TCO2 measured based on the whole core incubations.

    Second row: TCO2 production in anaerobic jar incubations.

    Third row: Sulfate reduction in carbon units (so converted 2:1).

Fourth row: relative contribution of SR to anaerobic respiration (so third row/second row * 100).

Fifth row: relative contribution of other anaerobic pathways (so (1- third row/second row)*100)

If this is correct, then your percentages do not make sense (if I estimate the relative contribution of SR to total anaerobic respiration, I get these values): 0.3/0.8 = 0.375, 4.7/8.7 = 0.54, 12.9/19.9=0.64, etc.

Also, I would suggest to maybe change the table: Keep the first two row, and then show the relative contributions of aerobic respiration – sulfate reduction – other anaerobic pathways.

*Author response: We have changed table 3 in accordance with this comment.*

**Response to major comments from Reviewer#2**

Sjøgaard et al. present an impressive data set exploring the effects of seawater flooding on carbon biogeochemistry in (1) agricultural and (2) a freshwater reedswamp with high temporal resolution over an extended 1 year time period (again, impressive). While the experiment itself was clearly well executed and the authors do a very nice job constructing a metabolic budget given the experimental constraints, the study design does not address the original hypotheses and cannot support the type of strong conclusions drawn by the authors.

> *Author response:*
>
> - *We thank Reviewer#2 for constructive criticism and are glad that he/she acknowledges that the study is based on a substantial experimental effort and a solid data-set that deserves to be published. We also acknowledge that Reviewer#2 feels very strongly that we are over-interpreting the data in some instances. In some instances we agree with the reviewer and have revised the text accordingly (see text below). In other cases, we argue our case and hope that the reviewer can see our point and accept our revisions.*

The authors have responded constructively to the original reviewer comments regarding methodology and literature, but have not addressed the key concerns from the original reviewers regarding the bulk of their interpretation of the data. Because there are no measurements from unflooded soils (C) or freshwater flooded soils (UC), there is no ambient condition with which to compare the effects of seawater flooding. Furthermore, there is no methodological control (i.e. cores from a seawater lagoon) to assess whether the laboratory handling (not seawater flooding) were responsible for the observed patterns. This does not doom the study, but severely restricts the type of solid conclusions that can be drawn to those regarding the effect of seawater on agricultural versus reedswamp soils only. In fact, the authors barely discuss the implications of antecedent conditions in the two sites (C & UC) (besides organic content) but it seems that the fact UC was a saturated (anaerobic) reedswamp is very important for framing and interpreting the results.

> *Author response:*
>
> - *The experimental set-up and vigorous sampling scheme prevented us from including additional experimental series in the experiment (for instance a series with unflooded soil, which would require a completely different sampling strategy or ambient controls from a seawater lagoon). However, the last author of the manuscript (TBV) has previously run similar, extensive long term experiments with different sediments from Odense Fjord – a lagoon very close to the sampling sites described in this manuscript (see Valdemarsen et al. 2014 (MEPS) and 2015 (Biogeosciences)) – we therefore have a kind of a methodological control as requested by the reviewer, and relatively good experience with interpreting results of this kind. We do not understand the critique in the last sentence – we very clearly state that the two soils under investigation have different origin and different characteristics, i.e. agricultural soil vs. reedswamp soils, and the whole point of the study is to see how the biogeochemistry develop in the different soils after flooding.*

While some speculation regarding impacts on broader C cycling is appropriate, the current manuscript extends speculation to strongly-worded conclusions regarding coastal carbon sinks that convey far more certainty than is appropriate. All three original referees objected to the author's conclusions as they are based on a fundamental misunderstanding of carbon-climate feedbacks and it is disappointing that the authors continue to emphasize this point over more evidence-based conclusions from their rich data set.

*Author response:*

- *We have removed or significantly down-played the strongly-worded conclusions regarding impacts of the current study in relation to global C-cycling, in order to accommodate the reviewer (see text below).*

Therefore the paper must undergo major revisions before it can be considered for publication. The following hypotheses and discussion pertinent to them must be significantly re-written:

H1 & H2: The authors cannot asses whether the origin or lability of organic matter plays any role in SOC degradation because lability is not measured and there is no control provided to show that recently flooded soils have a higher C loss that marine sediments under the same experimental conditions. While the chemical structure of material certainly impacts degradation, this is simply not measured in the current study and should not be the main focus.

*Author response:*

- *We disagree that the first hypothesis is not relevant and valid. To recap: H1 (page 2, line 34) states that total SOC degradation activity in soils after flooding depends on SOC content and lability.*

- *The reviewer seems to partially object to these hypotheses because 'organic matter lability' was not measured. We agree that organic matter lability was not measured directly, as in quantifying organic matter structure by biochemical analysis. However, the term lability in H1 should be perceived as a relative term related to the overall degradation patterns observed in the two soil types and compared to existing knowledge about carbon degradation processes in contemporary marine sediments. There is a whole body of literature regarding organic matter degradation in marine sediments supporting this use of the terms 'labile' and 'refractory' organic matter based on differences in degradation rates (e.g. Westrich and Berner 1984, Canfield 1994, Hedges and Keil 1995, Kristensen and Holmer 2001, Valdemarsen et al. 2009 etc. etc.).*

- *In the manuscript we compare the organic carbon degradation rates in a relatively organic rich soil (e.g. visibly rich in roots, organic debris) to a relatively organic poor soil (e.g. almost no visible organic debris). We find that the C-degradation rates are much higher in the organic rich soil and that in both soil types C-degradation appears to attenuate towards a much lower level. Ergo "We find that flooding of soils with differing soil organic content results in different, post-flooding degradation patterns, which can*

*only be explained by organic higher content and/or lability in the organic rich soil." We therefore confirm H1 (page 13, line 28).*

- *With regards to H2 (page 3, line 1) stating that "most SOC at the time of flooding will, due to its terrestrial origin, be non-degradable and hence preserved under the anoxic conditions formed after the flooding." We agree that this hypothesis was too boldly stated in the original version of the manuscript. We have revised it to: "a large proportion of SOC will be non-degradable due to the anoxic soil conditions forming after the flooding." Similarly we have rephrased the conclusion based on this hypothesis (page 13, line 13) to: "it appears that a large proportion of SOC is non-degradable under anoxic marine conditions and will essentially be preserved after flooding (hypothesis 2)." The reason for maintaining the hypothesis/conclusion is described below.*

- *Degradation in aquatic sediments follows exponential decay kinetics, meaning that rates decreases gradually with time as degradable organic C is depleted (e.g. Westrich and Berner 1984). In the current experiment we also found decreasing degradation rates in both soil types, although the temporal attenuation was much higher in the organic rich soil than in the organic poor soil. In both soil types, only about 6 to 7% of the initial soil organic C was degraded during the first year, and when considering the temporal decay patterns – and ESPECIALLY the DOC production close to 0 after 1 year - it is highly unlikely that the soil organic matter in any soil type would have been degraded within foreseeable time if the experiment had continued. We therefore conclude that: "a large proportion of SOC is non-degradable under anoxic marine conditions and will essentially be preserved after flooding." This statement follows the conclusions from a similar experiment lasting for 2 years, with 8 marine sediments collected in a nearby lagoon (Valdemarsen et al. 2014), where exponential decay patterns were explored in greater detail. Based on marine sediments showing similar decay patterns, generally <60-89% of organic matter present at the time of sampling appeared to be degradable under anoxic conditions. All though not directly comparable, we believe that a similar conclusion holds for the current case.*

H3: the study cannot support the key conclusions of the paper, namely that seawater flooding will preserve C, because the authors do not measure SOC degradation prior to flooding and their comparisons to agricultural rates rely on rates from disparate systems (some tropical, some global averages) measured largely in-situ (they therefor include autotrophic (root) respiration and are not comparable to the present study. Furthermore, respiration is meaningless if we don't know what gross primary productivity is. What is important is the net exchange of C, which is not addressed in this comparison with other studies. We can assume GPP it is 0 for the cores, so respiration is the only number in the equation (GPP-R=NEE) and the net exchange is negative (i.e. always out of the system). This is not the case for the other ag systems from the review. While it is clear that flooded soils preferentially preserve carbon in general, the authors should support this with outside literature as there results do not directly address this.

The conclusion that flooded site will constitute and immediate C-sink/negative carbon-climate feedback is highly objectionable and has great potential for misuse and misunderstanding. This is not a matter of data

interpretation or an over-extension of data, it is a incorrect. I implore the authors to consult the IPCC or Verified Carbon Standards (VCS) for finite definitions of the terms sink, stock, and source as they apply to greenhouse gas feedbacks. These terms are well established and clearly defined by the global change community.

Why the conclusion that flooded site will constitute and immediate C-sink/negative carbon-climate is inaccurate:

Agricultural soils can and do sequester C through the accumulation of crop residue (C sink), albeit at a lower rate than natural systems. As is the case with some drained agricultural land, it is also possible that SOC from reclaimed marine/intertidal/marsh sediments is still being lost to aerobic oxidation at a higher rate than crop residue is accumulating (C source). The authors present no evidence for either case as the antecedent (pre flood) condition was not measured. Thus there is no baseline to conclude how the direction of C flux has changed.

While it is not clear what the end-point of this particular coastal managed realignment is (subtidal or intertidal mudflat? subtidal seagrass bed? intertidal wetland?), the current study supposes it is a subtidal flat (always flooded, no vegetation) which can lead to 1 of 3 outcomes:

(1) assume ag land (C) was a C source (carbon emissions as respiration>crop C uptake) and flooding preserved C (100% preservation in 1 year= C stock) this is not the case in this study and even if it was a system simply cannot be a C sink/negative feedback unless primary production is removing C from the atmosphere. Prevented emissions do not equal negative carbon-climate feedbacks because C is NOT being removed from the atmosphere. Zero emission scenario.

(2) assume ag land (C) was a C sources, even at the measured 93% preservation in 1 year, the site is small C source and is a candidate for reduced emissions only, again not a sink, no negative climate feedback.

(3) ag land was a small sink (carbon uptake from crops > carbon emissions from ecosystem respiration) and flooding preserves 93% of the SOM. Flooding (without vegetation establishment) now makes the site a net source of C and thus there is a positive climate feedback. Furthermore, in the reedswamp (UC) soil, all indications are that saltwater increases respiration in freshwater anaerobic environments(Weston, Neubauer, and many other citations), this this represents the potential for positive feedback, not to mention the death of vegetation.

If the soils were vegetated (subtidal seagrass or intertidal wetland) then we have a candidate for a negative feedback.

I will reiterate that the data the authors have produced is interesting and impressive and should be published. It will be of great interest to the coastal community. As is, I have no qualms regarding the methods or data, only the interpretation. I encourage the authors to consider re-writing the hypotheses and discussion/conclusions in a way that emphasizes a direct connection to their results.

> *Author response:*
>
> - *We acknowledge that none of the three authors of the manuscript are experts in climate research, and we may therefore not be fully familiar with the terms and definitions*

*related to the work of IPCC. In the original version of the manuscript we concluded that since C-degradation appears to have almost ceased after 1 year of flooding and only 6-7% of initial soil organic C was degraded during the first year after flooding , then most of the soil organic C present at the time of flooding will most likely be permanently buried. We called this a "C-sink", and quantified it to constitute $9 \cdot 10^6$ kg organic C when considering the average organic content in soils at the study site. The basic assumptions for this calculation are OK – afterall if 93-94% of SOC remains after one year and net degradation of particulate SOC has ceased then the conclusion that "most SOC will be preserved" must be valid. However, we did one major error: we used the term C-sink without checking the correct use of the word "sink". According to the IPCC and the comments of Reviewer#2 a sink has to be permanent, i.e. be a lasting benefit to the global C-budget with annual benefits of more or less the same magnitude. In our case, however, the benefits related to our conclusion is temporary and only relates to the soil organic C being buried in the soils at the time of flooding. Our conclusion was never meant in relation to the long-term C-balance of the area – we agree that we do not have the data to support such a statement.*

- *To accommodate Reviewer#2's criticism we have made a number of changes to the manuscript:*

> *• Last sentence of the abstract was revised. Before it was: "On this basis we suggest that flooding of coastal soils through sea level rise or managed coastal realignment, will cause significant preservation of soil organic carbon and create an overall negative feedback on atmospheric carbon dioxide concentrations". The revised sentence reads: "On this basis we suggest that most of the organic carbon present in coastal soils exposed to flooding through sea level rise or managed coastal realignment will be permanently preserved."*

> *• We have deleted hypothesis 3 from the manuscript.*

> *• The headline of section 4.3 (page 12) was "Will newly flooded coastal habitats be hotspots for SOC burial?". It was changed to "Fate of SOC".*

> *• Refinement of argument (page 12, line 26): "The low final SOC degradation rates, and especially the very low final DOC production in both soil types, suggest that the majority of SOC present in soils at the time of flooding will be permanently buried…"*

> *• We have deleted the following sentence from section 4.3 (page 12) as it relates to the long-term C-balance of the area of which we have no knowledge: "Considering that terrestrial non-flooded vegetated soils generally have CO2 effluxes in the order of 0.1 to >1 mol m-2 d-1 (Chirinda et al., 2014; Fang and Moncrieff, 2001; Hursh et al., 2017; Rustad et al., 2001), which is much higher than measured in the flooded soils in this study (Table 4), it appears that flooding of coastal soils with seawater, due to either sea level rise*

*or mitigation techniques such as coastal realignment, will cause reduced CO2 efflux and a negative feedback on atmospheric CO2 concentrations."*

*• We have rephrased the concluding sentence of section 4.3. It is now clearly stated that we do not think our results indicate that the study area will constitute a C-sink. Concluding sentence now reads: " Hence flooding of coastal soils due to sea level rise or intentional flooding by managed realignment may lead to significant C-preservation. At Gyldensteen Strand SOC burial will be in the order of $48\pm6\cdot10^3$ kg SOC ha$^{-1}$ (average ± SEM, n = 30) when considering a detailed investigation of the soil characteristics down to 20 cm depth (T. Valdemarsen, unpublished results).Nevertheless this C-preservation does not constitute a permanent C-sink as it only relates to the SOC buried in the soils at the time of flooding."*

*• We have revised the concluding sentence in the manuscript. It was: "Hence this study suggests that in soils flooded with seawater the majority of SOC will be permanently preserved in comparison to non-flooded soils, therefore creating an overall negative feedback on atmospheric CO2 concentrations (hypothesis 3)." The revised sentence: "Hence this study suggests that in soils flooded with seawater the majority of SOC will be permanently preserved."*

Minor comments are included in attached PDF.

Referee Report:

bg-2016-417-referee-report.pdf

**Response to minor comments from reviewer#2 (comments extracted from pdf-file)**

**Page 1, line 7:** please provide a more descriptive phrasing that acknowledge manage coastal realignment applies to lands that are would naturally be within the range of tides

> **Author response:** Sentence was revised to: "…protect coastal areas lying below sea-level is intentional…"

**Page 1, line 11:** Give exact time line (e.g. number of days)

> **Author response:** Sentence was revised to: "We found rapid carbon degradation to TCO2 one day after experimental flooding and onwards and…"

**Page 1, line 14:** For the first year of the study

> **Author response:** It is mentioned in line 9 that the study lasted 1 year.

**Page 1, line 14:** This is misleading as no measure was made of carbon composition. Please remove or restate so it is clear the authors did not measure chemical composition of OM.

**Author response:** Sentence was rephrased to: "Organic carbon degradation decreased significantly after 6 months, indicating that most of the soil organic carbon was refractory towards microbial degradation under the anoxic conditions created in the soil after flooding.

**Page 1, line 16:** give duration of expt.

**Author response:** It is mentioned in line 9 that the study lasted 1 year.

**Page 1, line 18:** Reduced emissions do not equal a negative carbon-climate feedback. A negative feedback MUST include a mechanism for removing C from the atmosphere. At best, emissions are reduced (still non-zero), so the positive feedbacks are reduced, but the direction does not change.

**Author response:** Sentence was deleted.

**Page 1, line 22:** Not all will be familiar with "reclaimed" and "managed coastal realignment" terminology as they are region specific. Make sure to give a brief description of reclaimed as done below for MCR.

**Author response:** We do not agree that the term "Reclaimed coastal areas" needs to be explained in more detail.

**Page 1, line 26:** Citation? Gedan et al. 2012 etc.

**Author response:** The reference has been inserted, from 2011 though

**Page 2, line 2:** Review paper

**Author response:** The reference has been deleted

**Page 2, line 15:** This phrasing is very awkward... particularly "terminally oxidized"

**Author response:** and yet "terminally oxidized" is grammatically correct and frequently used in scientific texts. No revisions were made in response to this comment.

**Page 2, line 16:** Furthermore does not fit here. Consider re-organizing paragraph to discuss the basics of OM degradation common to all systems (enzyme hydrolysis, oxidation) and then contrast changes under anaerobic conditions. O2 is a terminal electron acceptor so flooding introduces a series of ALTERNATIVE terminal electron acceptors and relies more heavily on fermentation

**Author response:** "Furthermore" was deleted. Sentence was revised to: "Sulfate is abundant in seawater, and microbial sulfate reduction (SR) is therefore expected to become a major mineralization pathway in soils flooded with seawater (Sutton-Grier et al., 2011; Weston et al., 2011)."

**Page 2, line 28:** This negative feedback only occurs if flooding increases carbon uptake by the system (i.e. via the establishment of seagrass or emergent marsh). The scenario discussed in the paper is emissions reduction ONLY.

**Author response:** Sentence was revised to "Flooding of coastal soils by sea level rise and coastal realignment may therefore cause significant preservation of the SOC contained in the soils at the time of flooding."

**Page 3, line 15:** Depth? Subtidal or intertidal?

**Author response:** The tidal range in the area is only about ± 30 cm, so most of the lagoon is permanently subtidal with some of the areas closest to the coast being impacted by periodic exposure. Sentence was revised to: "…into a shallow and mostly subtidal marine lagoon."

**Page 10, line 25:** Wetland plants such as those from UC are aquatic plants, not terrestrial plants. Please revise terminology as vascular plants and plankton/algae

**Author response:** We have revised sentence. Sentence now read: "…stations was terrestrial and wetland plants such as grasses, reed and herbs rich in cellulose and lignified tissues…"

**Page 11, line 4:** The author's assumption that DOC is a proxy for microbial degradation of SOM is highly problematic. DOC can be generated by leaching of dry sediments (abiotic), change in ionic strength or other physicochemical changes, or by the death of microbial communities and leaching of cellular components. The authors must at least acknowledge that DOC can be a poor proxy for enzymatic hydrolysis, particularly as they did not include a control set of cores that were flooded only.

**Author response:** The first part of the sentence is referring to C-degradation in general and it is not appropriate to mention lysis and leaching here. We agree with the reviewer that initially DOC is not only produced as a result of degradation. Initially leaching as a result of flooding due to cell lysis etc. may be a significant DOC source, and this is mentioned in Page 11, line 10: "Part of this DOC may have leached to the porewater as a result of flooding…" Over the whole experimental duration, most DOC was produced as a result of microbial C-degradation.

**Page 11, line 15:** This result was attributed to increased flocculation of dissolved OM, not microbial processing

**Author response:** We have deleted reference to Ardon et al. 2016 here. We have added the sentence: "Valdemarsen et al. 2014 similarly observed gradually decreasing DOC production over 2 years in 8 different sediment types from Odense Fjord, indicating gradual depletion of degradable organic matter despite high sediment organic content and abundance of energetically favorable electron acceptors."

**Page 11, line 25:** All microbes require a terminal electron acceptor. Please revise text throughout.

**Author response:** Sentence was revised to: "indicating that microbes oxidizing DOC to CO2 adapt slower to flooded conditions than fermenting and hydrolyzing microbes."

**Page 11, line 26:** No evidence it is hydrolysis not abiotic process.

**Author response:** Sentence was revised to: "Similar cases of initial DOC-production due to leaching and/or substrate hydrolysis.."

**Page 12, line 23:** This Neubauer study would be analogous to the UC site (reedswamp) in the present study and Neubauer and the present study show that the introduction of seawater increases the metabolism of soil carbon. Neubauer showed that saltwater intrusion would enhance SOC loss, not store more carbon.

**Author response:** We are referring to the conclusion in the Neubauer paper about the effect long term exposure to seawater on SOC degradation rates.

**Page 12, line 26:** These numbers include autotrophic respiration as well as heterotrophic and are not useful for comparison unless the soil carbon stock is taken into account (i.e. what % of soil C stock is lost to heterotrophic respiration.)

**Author response:** line and table 4 were deleted. Comment no longer relevant.

**Page 12, line 29:** Please consider that many agricultural soils do sequester C (remove carbon from the atmosphere via photosynthesis and preserve it as SOM), albeit at a lower rate that natural systems. The authors are only showing C loss from the sediments, which will be a C source unless colonized by some primary producer that will add C.

**Page 13, line 1:** This is not a sink. This is a stock.

**Author response:** sentence was revised

[revised manuscript text omitted]

Kamilla 5/7/2017 13:59

Thomas Bruun Valdem…, 3/7/2017 10:30

Thomas Bruun Valdem…, 3/7/2017 10:30

Thomas Bruun Valdem…, 3/7/2017 10:32

Thomas Bruun Valdem…, 3/7/2017 10:34

Thomas Bruun Valdem…, 3/7/2017 10:34

Thomas Bruun Valdem…, 3/7/2017 10:34

Thomas Bruun Valdem…, 3/7/2017 10:37

Thomas Bruun Valdem…, 3/7/2017 10:39

Thomas Bruun Valdem…, 3/7/2017 10:39

Thomas Bruun Valdem…, 3/7/2017 10:40

Thomas Bruun Valdem…, 3/7/2017 10:41

Thomas Bruun Valdem…, 3/7/2017 10:41

Kamilla 5/7/2017 14:03

Thomas Bruun Valdem…, 4/7/2017 11:15

Thomas Bruun Valdem…, 4/7/2017 11:15

Thomas Bruun Valdem…, 4/7/2017 11:16

Thomas Bruun Valdem…, 3/7/2017 12:57

Thomas Bruun Valdem…, 3/7/2017 12:58

Thomas Bruun Valdem…, 3/7/2017 12:58

[revised manuscript text omitted]